



Systematic Characterization and Fluorescence Threshold Strategies for the Wideband Integrated
Bioaerosol Sensor (WIBS) Using Size-Resolved Biological and Interfering Particles
NICOLE SAVAGE[1], Christine Krentz[1], Tobias Könemann[2], Taewon T. Han[3], Gediminas
Mainelis[3], Christopher Pöhlker[2], John A. Huffman[1]
[1] *University of Denver, Department of Chemistry and Biochemistry, Denver, USA*
[2] *Max Planck Institute for Chemistry, Multiphase Chemistry and Biogeochemistry Departments,*
*Mainz, Germany*
[3] *Rutgers, The State University of New Jersey, Department of Environmental Science, New*
*Jersey, USA*
**Abstract**
Atmospheric particles of biological origin, also referred to as bioaerosols or primary biological
aerosol particles (PBAP), are important to various human health and environmental systems.
There has been a recent steep increase in the frequency of published studies utilizing commercial
instrumentation based on ultraviolet laser/light-induced fluorescence (UV-LIF), such as the
WIBS (wideband integrated bioaerosol sensor) or UV-APS (ultraviolet aerodynamic particle
sizer), for bioaerosol detection both outdoors and in the built environment. Significant work over
several decades supported the development of the general technologies, but efforts to
systematically characterize the operation of new commercial sensors has remained lacking.
Specifically, there have been gaps in the understanding of how different classes of biological and
non-biological particles can influence the detection ability of LIF-instrumentation. Here we
present a systematic characterization of the WIBS-4A instrument using 69 types of aerosol
materials, including a representative list of pollen, fungal spores, and bacteria as well as the most
important groups of non-biological materials reported to exhibit interfering fluorescent
properties. Broad separation can be seen between the biological and non-biological particles
directly using the five WIBS output parameters and by taking advantage of the particle
classification analysis introduced by Perring et al. (2015). We highlight the importance that
particle size plays on observed fluorescence properties and thus in the Perring-style particle
classification. We also discuss several particle analysis strategies, including the commonly used
fluorescence threshold defined as the mean instrument background (forced trigger; FT) plus 3
standard deviations ($\sigma$) of the measurement. Changing the particle fluorescence threshold was
shown to have a significant impact on fluorescence fraction and particle type classification. We
conclude that raising the fluorescence threshold from FT + 3$\sigma$ to FT + 9$\sigma$ does little to reduce the
relative fraction of biological material considered fluorescent, but can significantly reduce the
interference from mineral dust and other non-biological aerosols. We discuss examples of highly
fluorescent interfering particles, such as brown carbon, diesel soot, and cotton fibers, and how
these may impact WIBS analysis and data interpretation in various indoor and outdoor
environments. A comprehensive online supplement is provided, which includes size distributions
broken down by fluorescent particle type for all 69 aerosol materials and comparing two
threshold strategies. Lastly, the study was designed to propose analysis strategies that may be
useful to the broader community of UV-LIF instrumentation users in order to promote deeper



discussions about how best to continue improving UV-LIF instrumentation and analysis
strategies.

## 1. Introduction

Biological material emitted into the atmosphere from biogenic sources on terrestrial and
marine surfaces can play important roles in the health of many living systems and may influence
diverse environmental processes (Cox and Wathes, 1995;Pöschl, 2005;Després et al.,
2012;Fröhlich -Nowoisky et al., 2016). Bioaerosol exposure has been an increasingly important
component of recent interest, motivated by studies linking airborne biological agents and adverse
health effects in both indoor and occupational environments (Douwes et al., 2003). Bioaerosols
may also impact the environment by acting as giant cloud condensation nuclei (GCCN) or ice
nuclei (IN), having an effect on cloud formation and precipitation (Ariya et al., 2009;Delort et
al., 2010;Möhler et al., 2007;Morris et al., 2004).  Biological material emitted into the
atmosphere is commonly referred to as Primary Biological Aerosol Particles (PBAP) or
bioaerosols. PBAP can include whole microorganisms, such as bacteria and viruses, reproductive
entities (fungal spores and pollen) and small fragments of any larger biological material, such as
leaves, vegetative detritus, fungal hyphae, or biopolymers, and can represent living, dead,
dormant, pathogenic, allergenic, or biologically inert material (Després et al., 2012). PBAP often
represent a large fraction of supermicron aerosol, for example up to 65% by mass in pristine
tropical forests, and may also be present in high enough concentrations at submicron sizes to
influence aerosol properties (Jaenicke, 2005;Penner, 1994;Pöschl et al., 2010).
Until recently the understanding of physical and chemical processes involving bioaerosols
has been limited due to a lack of instrumentation capable of characterizing particles with
sufficient time and size resolution (Huffman and Santarpia, 2017). The majority of bioaerosol
analysis historically utilized microscopy or cultivation-based techniques. Both are time-
consuming, relatively costly and cannot be utilized for real-time analysis (Griffiths and
Decosemo, 1994;Agranovski et al., 2004). Cultivation techniques can provide information about
properties of the culturable fraction of the aerosol (e.g. bacterial and fungal spores), but can
greatly underestimate the diversity and abundance of bioaerosols because the vast majority of
microorganism species are not culturable (Amann et al., 1995;Chi and Li, 2007;Heidelberg et al.,
1997). Further, because culture-based methods cannot detect non-viable bioaerosols, information
about their chemical properties and allergenicity has been poorly understood.
In recent years, advancements in the chemical and physical detection of bioaerosols have
enabled the development of rapid and cost-effective techniques for the real-time characterization
and quantification of airborne biological particles (Ho, 2002;Hairston et al., 1997;Huffman and
Santarpia, 2017;Sodeau and O'Connor, 2016). One important technique is based on ultraviolet
laser/light-induced fluorescence (UV-LIF), originally developed by military research
communities for the rapid detection of bio-warfare agents (BWA) (e.g. Hill et al., 2001;Hill et
al., 1999a;Pinnick et al., 1995). More recently, UV-LIF instrumentation has been
commercialized for application toward civilian research in fields related to atmospheric and
exposure science. The two most commonly applied commercial UV-LIF bioaerosol sensors are
the wideband integrated bioaerosol sensor (WIBS; University of Hertfordshire, Hertfordshire,
UK, now licensed to Droplet Measurement Technologies, Longmont, CO, USA), and the
ultraviolet aerodynamic particle sizer (UV-APS; licensed to TSI, Shoreview, MN, USA). Both



sensors utilize pulsed ultraviolet light to excite fluorescence from individual particles in a real-
time system. The wavelengths of excitation and emission were originally chosen to detect
biological fluorophores assumed to be widely present in airborne microorganisms (e.g.
tryptophan-containing proteins, NAD(P)H co-enzymes, or riboflavin) (Pöhlker et al., 2012).
Significant work was done by military groups to optimize pre-commercial sensor performance
toward the goal of alerting for the presence of biological warfare agents such as anthrax spores.
The primary objective from this perspective is to positively identify BWAs without being
distracted by false-positive signals from fluorescent particles in the surrounding natural
environment (Primmerman, 2000). From the perspective of basic atmospheric science, however,
the measurement goal is often to quantify bioaerosol concentrations in a given environment. So,
to a coarse level of discrimination, BWA-detection communities aim to ignore most of what the
atmospheric science community seeks to detect. Researchers on such military-funded teams also
have often not been able to publish their work in formats openly accessible to civilian
researchers, so scientific literature is lean on information that can help UV-LIF users operate and
interpret their results effectively. Early UV-LIF bioaerosol instruments have been in use for two
decades and commercial instruments built on similar concepts are emerging and becoming
widely used by scientists in many disciplines. In some cases, however, papers are published with
minimal consideration of complexities of the UV-LIF data. This study presents a detailed
discussion of several important variables specific to WIBS data interpretation, but that can apply
broadly to operation and analysis of many similar UV-LIF instruments.
The commercially available WIBS instrument has become one of the most commonly
applied instrument toward the detection and characterization of bioaerosol particles in both
outdoor and indoor environments. As will be discussed in more detail, the instrument utilizes two
wavelengths of excitation (280 nm and 370 nm), the second of which is close to the one
wavelength utilized by the UV-APS (355 nm). Both the WIBS and UV-APS, in various version
updates, have been applied to many types of studies regarding outdoor aerosol characterization.
For example they have been important instruments: in the study of ice nuclei (Huffman et al.,
2013;Mason et al., 2015;Twohy et al., 2016), toward the understanding of outdoor fungal spore
concentrations (Gosselin et al., 2016;Saari et al., 2015a;O'Connor et al., 2015b), to investigate
the concentration and properties of bioaerosols from long-range transport (Hallar et al., 2011), in
tropical aerosol (Gabey et al., 2010;Whitehead et al., 2010;Huffman et al., 2012;Valsan et al.,
2016;Whitehead et al., 2016), in urban aerosol (Huffman et al., 2010;Saari et al., 2015b;Yu et al.,
2016), from composting centers (O'Connor et al., 2015), at high altitude (Crawford et al.,
2016;Gabey et al., 2013;Perring et al., 2015;Ziemba et al., 2016), and in many other
environments (Healy et al., 2014;Li et al., 2016;O'Connor et al., 2015a). The same
instrumentation has been utilized for a number of studies involving the built, or indoor,
environment as well (Wu et al., 2016). As a limited set of examples, these instruments have been
critical components in the study of bioaerosols in the hospital environment (Lavoie et al.,
2015;Handorean et al., 2015) and to study the emission rates of biological particles directly from
humans (Bhangar et al., 2016) in school classrooms (Bhangar et al., 2014), and in offices (Xie et
al., 2017).
Despite the numerous and continually growing list of studies that utilize commercial UV-LIF
instrumentation, only a handful of studies have published results from laboratory work
characterizing the operation or analysis of the instruments in detail. For example, Kananni et al.
(2007, 2008, 2009) and Agranovski et al. (2003, 2004, 2005) presented several examples of UV-





APS operation with respect to bio-fluorophores and biological particles. Healy et al. (2012)
provided an overview of fifteen spore and pollen species analyzed by the WIBS, and Toprak and
Schnaiter (2013) discussed the separation of dust from ambient fluorescent aerosol by applying a
simple screen of any particles that exhibited fluorescence in one specific fluorescent channel.
Hernandez et al. (2016) presented a summary of more than 50 pure cultures of bacteria, fungal
spores, and pollen species analyzed by the WIBS and with respect to fluorescent particle type.
Fluorescent particles observed in the atmosphere have frequently been used as a lower-limit
proxy for biological particles (e.g. Huffman et al. 2010), however it is well known that a number
of key particle types of non-biological origin can fluoresce.  For example, certain examples of
soot, humic and fulvic acids, mineral dusts, and aged organic aerosols can exhibit fluorescent
properties, and the effects that these play in the interpretation of WIBS data is unclear (Bones et
al., 2010; Gabey et al., 2011; Lee et al., 2013; Pöhlker et al., 2012; Sivaprakasam et al., 2004).

The simplest level of analysis of WIBS data is to provide the number of particles that exceed
the minimum detectable threshold in each of the three fluorescence categories. Many papers on
ambient particle observations have been written using this data analysis strategy with both the
WIBS and UV-APS data. Such analyses are useful and can provide an important first layer of
discrimination by fluorescence. To provide more complicated discrimination as a function of
observed fluorescence intensity, however, brings associated analysis and computing challenges,
i.e. users often must write data analysis code themselves and processing large data sets can push
the limits of standard laboratory computers.  Discriminating based on fluorescence intensity also
requires more detailed investigations into the strategy by which fluorescent thresholds can be
applied to define whether a particle is considered fluorescent.  Additionally, relatively little
attention has been given to the optical properties of non-biological particles interrogated by the
WIBS and to optimize how best to systematically discriminate between biological aerosol of
interest and materials interfering with those measurements.

Here we present a comprehensive and systematic laboratory study of WIBS data in order to
aid the operation and data interpretation of commercially available UV-LIF instrumentation. This
work presents 69 types of aerosol materials, including key biological and non-biological
particles, interrogated by the WIBS-4A and shows the relationship of fluorescent intensity and
resultant particle type as a function of particle size and asymmetry. A discussion of thresholding
strategy is given, with emphasis on how varying strategies can influence characterization of
fluorescent properties and either under- or over-prediction of fluorescent biological particle
concentration.

## 2. WIBS Instrumentation

### 2.1 Instrument Design and Operation

The WIBS (Droplet Measurement Technologies; Longmont, Colorado) uses light scattering
and fluorescence spectroscopy to detect, size, and characterize the properties of interrogated
aerosols on a single particle basis (instrument model 4A utilized here). Air is drawn into the
instrument at a flow rate of 0.3 L/min and surrounded by a filtered sheath flow of 2.2 L/min. The
aerosol sample flow is then directed through an intersecting a 635 nm, continuous wave (cw)
diode laser, which produces elastic scattering measured in both the forward and side directions.
Particle sizing in the range of approximately 0.5 µm to 20 µm is detected by the magnitude of



the electrical pulse detected by a photomultiplier tube (PMT) located at 90 degrees from the laser
beam. Particles whose measured cw laser-scattering intensity (particle size) exceed user-
determined trigger thresholds will trigger two xenon flash lamps (Xe1 and Xe2) to fire in
sequence, approximately 10 microseconds apart. The two pulses are optically filtered to emit at
280 nm and 370 nm, respectively. Fluorescence emitted by a given particle after each excitation
pulse is detected simultaneously using two PMT detectors. The first PMT is optically filtered to
detect the total intensity of fluorescence in the range 310-400 nm and the second PMT in the
range 420-650 nm. So for every particle that triggers xenon lamp flashes, Xe1 produces a signal
in the FL1 (310-400 nm) and FL2 (420-650 nm) channels, whereas the Xe2 produces only a
signal in the FL3 (420-650 nm) channel because elastic scatter from the Xe2 flash saturates the
first PMT. The WIBS-4A has two user defined trigger thresholds, T1 and T2 that define which
data will be recorded. Particles producing a scattering pulse from the cw laser that is below the
T1 threshold will not be recorded. This enables the user to reduce data collection during
experiments with high concentrations of small particles. Particles whose scattering pulse exceeds
the T2 threshold will trigger xenon flash lamp pulses for interrogation of fluorescence. Note that
the triggering thresholds mentioned here are fundamentally different from the analysis thresholds
that will be discussed in detail later.

Forward-scattered light is detected using a quadrant PMT. The detected light intensity in
each quadrant are combined using Equation 1 into an asymmetry factor (AF), where $k$ is an
instrument defined constant, $E$ is the mean intensity measured over the entire PMT, and $E_i$ is the
intensity measured at the $i^{th}$ quadrant (Gabey et al., 2010).
$$AF = \frac{k\left(\sum_{i=1}^{n}(E-E_i)^2\right)^{1/2}}{E} \quad (1)$$

This parameter relates to a rough estimate of the sphericity of an individual particle by
measuring the difference of light intensity scattered into each of the four quadrants. A perfectly
spherical particle would theoretically exhibit an AF value of 0, whereas larger AF values greater
than 0 and less than 100, indicate rod-like particles (Kaye et al., 1991;Gabey et al., 2010;Kaye et
al., 2005). It is important to note that this parameter is not rigorously a shape factor like used in
other aerosol calculations (DeCarlo et al., 2004;Zelenyuk et al., 2006) and only very roughly
relates a measure of particle sphericity.
**2.2 WIBS Calibration**

Particle sizing within the instrument was calibrated periodically by aerosolizing several sizes
of non-fluorescent polystyrene latex spheres (PSLs; Polysciences, Inc., Pennsylvania), including
0.51 µm (part number 07307), 0.99 µm. (07310), 1.93 µm (19814), 3.0 µm (17134), and 4.52 µm
(17135). A histogram of signal intensity was plotted separately for each PSL, and the peak of a
Gaussian fit to those data was then plotted versus the physical diameter of the PSL. A second
degree polynomial fit was used to generate an equation in order to calibrate side scatter values
into size.

Fluorescence intensity in each WIBS channel was calibrated using 2.0 µm Green (G0200),
2.1 µm Blue (B0200), and 2.0 µm Red (R0200) fluorescent PSLs (Thermo-Scientific,
Sunnyvale, California). For each particle type, a histogram of the fluorescence intensity signal in



each channel was fitted with a Gaussian function, and the median intensity was recorded.
Periodic checks were performed using the same stock bottles of the PSLs in order to verify that
mean fluorescence intensity of each had not shifted more than one standard deviation between
particle sample types (Table 1). The particle fluorescence standards used present limitations due
to variations in fluorescence intensity between stocks of particles and due to fluorophore
degradation over time. To improve reliability between instruments, stable fluorescence standards
and calibration procedures (e.g. Robinson et al., 2017) will be important.
Voltage gain settings for the three PMTs that produce sizing, fluorescence, and AF values,
respectively, significantly impact measured intensity values and are recorded here for rough
comparison of calibrations and analyses to other instruments. The voltage settings used for all
data presented here were set according to manufacturer specifications and are as follows: $PMT_1$
(AF) 400 V, $PMT_2$ (particle sizing and FL1 emission) 450 mV, and $PMT_3$ (FL2, FL3 emission)
732 mV.

### 225   2.3 WIBS Data Analysis

An individual particle is considered to be fluorescent in any one of the three fluorescence
channels (FL1, FL2, or FL3) when its fluorescence emission intensity exceeds a given baseline
threshold. The baseline fluorescence can be determined by a number of strategies, but commonly
has been determined by measuring the observed fluorescence in each channel when the xenon
lamps are fired into the optical chamber when devoid of particles. This is referred to as the
"forced trigger" (FT) process, because the xenon lamp firing is not triggered by the presence of a
particle. The instrument background is also dependent on the intensity and orientation of Xe
lamps, voltage gains of PMTs, quality of PMTs based on production batch, orientation of optical
components i.e. mirrors in the optical chamber, etc. As a result of these factors, the background
or baseline of a given instrument is unique and cannot been used as a universal threshold. All
threshold values used in this study can are listed in supplementary Table S1. Fluorescence
intensity in each channel is recorded at an approximate FT rate of one value per second for a
user-defined time period, typically 30-120 seconds. The baseline threshold in each channel has
typically been determined as the average plus 3x the standard deviation ($\sigma$) of forced trigger
fluorescence intensity measurement (Gabey et al., 2010), however alternative applications of the
fluorescence threshold will be discussed. Particles exhibiting fluorescence intensity lower than
the threshold value in each of the three channels are considered to be non-fluorescent. The
emission of fluorescence from any one channel is essentially independent of the emission in the
other two channels. The pattern of fluorescence measured allows particles to be categorized into
7 fluorescent particle types (A, B, C, AB, AC, BC, or ABC) as depicted in Figure 1, or as
completely non-fluorescent (Perring et al., 2015).
Other threshold strategies have also been proposed and will be discussed. For example,
Wright et al. (2014) used set fluorescence intensity value boundaries rather than using the
standard Gabey et al. (2010) definition that applies a threshold as a function of observed
background fluorescence. The Wright et al. (2014) study proposed five separate categories of
fluorescent particles (FP1 through FP5). Each definition was determined by selecting criteria for
excitation-emission boundaries and observing the empirical distribution of particles in a 3-
dimensional space (FL1 vs. FL2 vs. FL3). For the study reported here, only the FP3 definition
was used for comparison, because Wright et al. (2014) postulated the category as being enriched



with fungal spores during their ambient study and because they observed that these particles
scaled more tightly with observed ice nucleating particles. The authors classified a particle in the
FP3 category if the fluorescence intensity in FL1 > 1900 arbitrary units (a.u) and between 0-500
a.u for each FL2 and FL3.

### 3. Materials and methods

#### 3.1 Aerosol Materials

*3.1.1 Table of materials*

All materials utilized, including the vendors and sources from where they were acquired,
have been listed in supplemental Table S1, organized into broad particle type groups: biological
material (fungal spores, pollen, bacteria, and biofluorophores) and non-biological material (dust,
humic-like substances or HULIS, polycyclic aromatic hydrocarbons or PAHs, combustion soot
and smoke, and miscellaneous non-biological materials). Combustion soot and smoke are
grouped into one set of particles analyzed and are hereafter referred to as "soot" samples.

*3.1.2 Brown carbon synthesis*

Three different brown carbon solutions were synthesized using procedures described by
Powelson et al. (2014): (Rxn 1) methylglyoxal + glycine, (Rxn 2) glycolaldehyde +
methylamine, and (Rxn 3) glyoxal + ammonium sulfate. Reactions conditions were reported
previously, so only specific concentration and volumes used here are described. All solutions
described are aqueous and were dissolved into 18.2 MΩ water (Millipore Sigma; Denver, CO).
For reaction 1, 25.0 mL of 0.5 M methylglyoxal solution was mixed with 25 mL of 0.5 M
glycine solution. For reaction 2, 5.0 mL of 0.5 M glyoxal trimer dihydrate solution was mixed
with 5.0 mL of 0.5 M ammonium sulfate solution. For reaction 3, 10.0 mL of 0.5 M
glycolaldehyde solution was mixed with 10.0 mL of 0.5 M methylamine solution. The pH of the
solutions was adjusted to approximately pH 4 by adding 1 M oxalic acid in order for the reaction
to follow the appropriate chemical mechanism (Powelson et al., 2014). The solutions were
covered with aluminum foil and stirred at room temperature for 8 days, 4 days, and 4 days, for
reactions 1, 2, and 3, respectively. Solutions were aerosolized via the liquid aerosolization
method described in Section 3.2.4.

#### 3.2 Aerosolization Methods

*3.2.1 Fungal spore growth and aerosolization*

Fungal cultures were inoculated onto sterile, disposable polystyrene plates (Carolina,
Charlotte, NC) filled with agar growth media consisting of malt extract medium mixed with
0.04 M of streptomycin sulfate salt (S6501, Sigma-Aldrich) to suppress bacterial colony growth.
Inoculated plates were allowed to mature and were kept in a sealed Plexiglas box for 3-5 weeks
until aerosolized. Air conditions in the box were monitored periodically and were consistently
25-27 °C and 70% relative humidity.
Fungal cultures were aerosolized inside an environmental chamber constructed from a re-
purposed home fish tank (Aqueon Glass Aquarium, 5237965). The chamber has glass panels



with dimensions 20.5 L x 10.25 H x 12.5 W in (supplemental Fig. S1). Soft rubber beading seals
the top panel to the walls, allowing isolation of air and particles within the chamber. Two tubes
are connected to the lid. The first delivers pressurized and particle-free air through a bulkhead
connection, oriented by plastic tubing (Loc-Line Coolant Hose, 0.64 inch outer diameter) and a
flat nozzle. The second tube connects 0.75 inch internal diameter conductive tubing (Simolex
Rubber Corp., Plymouth, MI) for aspiration of fungal aerosol, passing it through a bulkhead
fitting and into tubing directed toward the WIBS. Aspiration tubing is oriented such that a gentle
90-degree bend brings aerosol up vertically through the top panel.
For each experiment, an agar plate with a mature fungal colony was sealed inside the
chamber. The air delivery nozzle was positioned so that a blade of air was allowed to approach
the top of the spore colony at a shallow angle in order to eject spores into an approximately
horizontal trajectory. The sample collection tube was positioned immediately past the fungal
plate to aspirate aerosolized fungal particles. Filtered room air was delivered by a pump through
the aerosolizing flow at approximately 9 – 15 L/min, varied within each experiment to optimize
measured spore concentration. Sample flow was 0.3 L/min into the WIBS and excess input flow
was balanced by outlet through a particle filter connected through a bulkhead on the top plate.
Two additional rubber septa in the top plate allow the user to manipulate two narrow metal
rods to move the agar plate once spores were depleted from a given region of the colony. After
each spore experiment, the chamber and tubing was evacuated by pumping for 15 minutes, and
all interior surfaces were cleaned with isopropanol to avoid contamination between samples.
*3.2.2 Bacterial growth and aerosolization*
All bacteria were cultured in nutrient broth (Becton, Dickinson and Company, Sparks, MD)
for 18 hours in a shaking incubator at 30°C for *Bacillus atrophaeus* (ATCC 49337, American
Type Culture Collection, MD), 37°C for *Escherichia coli* (ATCC 15597), and 26°C
*Pseudomonas fluorescens* (ATCC 13525).  Bacterial cells were harvested by centrifugation at
7000 rpm (6140 g) for 5 min at 4°C (BR4, Jouan Inc., Winchester, VA) and washed 4 times with
autoclave-sterilized deionized water (Millipore Corp., Billerica, MA) to remove growth media.
The final liquid suspension was diluted with sterile deionized water, transferred to a
polycarbonate jar and aerosolized using a three jet Collison nebulizer (BGI Inc., Waltham, MA)
operated at 5 L/min (pressure of 12 psi). The polycarbonate jar was used to minimize damage to
bacteria during aerosolization (Zhen et al., 2014 ) . The tested airborne cell concentration was
about ~$10^5$ cells/Liter as determined by an optical particle counter (model 1.108, Grimm
Technologies Inc., Douglasville, GA). Bacterial aerosolization took place in an experimental
system containing a flow control system, a particle generation system, and an air-particle mixing
system introducing filtered air at 61 L/min as described by Han et al. (2015).
*3.2.3 Powder aerosolization*
Dry powders were aerosolized by mechanically agitating material by one of several methods
mentioned below and passing filtered air across a vial containing the powder. For each method,
approximately 2.5-5.0 g of sample was placed in a 10 mL glass vial. For most samples (method
P1), a stir bar was added, and the vial was placed on a magnetic stir plate. Two tubes were
connected through the lid of the vial. The first tube connected a filter, allowing particle-free air





to enter the vessel. The second tube connected the vial through approximately 33 cm of
conductive tubing (0.25 in inner diam.) to the WIBS for sample collection.
The setup was modified (method P2) for a small subset of samples whose solid powder was
sufficiently fine to produce high number concentrations of submicron aerosol particles that could
risk coating the internal flow path and damaging optical components of the instrument. In this
case, the same small vial with powder and stir bar was placed in a larger reservoir (~0.5 L), but
without vial lid. The lid of the larger reservoir was connected to filtered air input and an output
connection to the instrument. The additional container volume allowed for greater dilution of
aerosol before sampling into the instrument.
Some powder samples produced consistent aerosol number concentration even without
stirring. For these samples, $2.5 - 5.0$ g of material was placed in a small glass vial and set under a
laboratory fume hood (method P3). Conductive tubing was held in place at the opening of the
vial using a clamp, and the opposite end was connected to the instrument with a flow rate of 0.3
L/min. The vial was tapped by hand or with a hand tool, physically agitating the material and
aerosolizing the powder.
*3.2.4 Liquid aerosolization*
Disposable, plastic medical nebulizers (Allied Healthcare, St. Louis, MO) were used to
aerosolize liquid solutions and suspensions. Each nebulizer contains a reservoir where the
solution is held. Pressurized air is delivered through a capillary opening on the side, reducing
static pressure and, as a result, drawing fluid into the tube. The fluid is broken up by the air jet
into a dispersion of droplets, where most of the droplets are blown onto the internal wall of the
reservoir, and droplets remaining aloft are entrained into the sample stream. Output from the
medical nebulizer was connected to a dilution chamber (aluminum enclosure, 0.5 L), allowing
the droplets to evaporate in the system before particles enter the instrument for detection.
*3.2.5 Smoke generation*
Wood and cigarette smoke samples were aerosolized through combustion. Each sample was
ignited separately using a personal butane lighter while held underneath a laboratory fume hood.
Once the flame from the combusting sample was naturally extinguished, the smoldering sample
was waved at a height ~5 cm above the WIBS inlet for 3– 5 minutes during sampling.

**3.3 Pollen microscopy**

Pollen samples were aerosolized using the dry powder vial (P1, P2) and tapping (P3)
methods detailed above. Samples were also collected by impaction onto a glass microscope slide
for visual analysis using a home-built, single-stage impactor with $D_{50}$ cut ~0.5 µm at flow-rate
1.2 L min$^{-1}$. Pollen were analyzed using an optical microscope (VWR model 89404-886) with a
40x objective lens. Images were collected with an AmScope complementary metal-oxide
semiconductor camera (model MU800, 8 megapixels).

**4. Results**
**4.1 Broad separation of particle types**



The WIBS is routinely used as an optical particle counter applied to the detection and
characterization of fluorescent biological aerosol particles (FBAP). Each interrogated particle
provides five discreet pieces of information: fluorescence emission intensity in each of the 3
detection channels (FL1, FL2, and FL3), particle size, and particle asymmetry. Thus, a thorough
summary of data from aerosolized particles would require the ability to show statistical
distributions in five dimensions. As a simple, first-order representation of the most basic
summary of the 69 particle types analyzed, Figure 2 and Table 1 show median values for each of
the five data parameters plotted in three plot styles (columns of panels in Fig. 2).
For the sake of WIBS analysis, each pollen type was broken into two size categories, because
it was observed that most pollen species exhibited two distinct size modes. The largest size mode
peaked above 10 µm in all cases and often saturated the sizing detector (see also fraction of
particles that saturated particle detector for each fluorescence channel in Table 2). This was
interpreted to be intact pollen. A broad mode also usually appeared at smaller particle diameters
for some pollen species, suggesting that pollen grains had ruptured during dry storage or through
the mechanical agitation process. This hypothesis was supported by optical microscopy through
which a mixture of intact pollen grains and ruptured fragments were observed (Fig. S2). For the
purposes of this investigation, the two modes were separated at the minimum point between
modes in order to observe optical properties of the intact pollen and pollen fragments separately.
The list number for each pollen (Tables 2, S1) is consistent for the intact and fragmented species,
though not all pollen exhibited obvious pollen fragments.
The WIBS was developed primarily to discriminate biological from non-biological particles,
and the three fluorescence channels broadly facilitate this separation. Biological particles, i.e.
pollen, fungal spores, and bacteria (top row of Fig. 2), each show strong median fluorescence
signal in at least one of the three channels. In general, all fungal spores sampled (blue dots) show
fluorescence in the FL1 channel with lower median emission in FL2 and FL3 channels. Both the
fragmented (pink dots) and intact (orange dots) size fractions of pollen particles showed high
median fluorescence emission intensity in all channels, varying by species and strongly as a
function of particle size. The three bacterial species sampled (green dots) showed intermediate
median fluorescence emission in the FL1 channel and very low median intensity in either of the
other two channels.  To support the understanding of whole biological particles, pure molecular
components common to biological material were aerosolized separately and are shown as the
second row of Figure 2.  Each of the biofluorophores chosen shows relatively high median
fluorescence intensity, again varying as a function of size. Key biofluorophores such as NAD,
riboflavin, tryptophan, and tyrosine are individually labeled in Figure 2d. Supermicron particles
of these pure materials would not be expected in a real-world environment, but are present as
dilute components of complex biological material and are useful here for comparison. In general,
the spectral properties summarized here match well with fluorescence excitation emission
matrices (EEMs) presented by Pöhlker et al. (2012;2013)
In contrast to the particles of biological origin, a variety of non-biological particles were
aerosolized in order to elucidate important trends and possible interferences. The majority of
non-biological particles shown in the bottom row of Figure 2 show little to no median
fluorescence in each channel and are therefore difficult to differentiate from one another in the
figure. For example, Figure 2g (lower left) shows the median fluorescence intensity of 6 different
groups of particle types (33 total dots), but almost all overlap at the same point at the graph
origin. The exceptions to this trend include the PAHs (blue dots), miscellaneous particles (green)
and several types of combustion soot (black dots). The fluorescent properties of PAHs are well-
known in both basic chemical literature and as observed in the atmosphere (Niessner and Krupp,
1991;Finlayson-Pitts and Pitts, November 1999;Panne et al., 2000;Slowik et al., 2007).  PAHs
can be produced by a number of anthropogenic sources and are emitted in the exhaust from
vehicles and other combustion sources as well as from biomass burning (Aizawa and Kosaka,
2010, 2008;Abdel-Shafy and Mansour, 2016;Lv et al., 2016). PAHs alone exhibit high
fluorescence quantum yields (Pöhlker et al., 2012;Mercier et al., 2013), but as pure materials are
not usually present in high concentrations at sizes large enough (>0.8 µm) to be detected by the
WIBS. Highly fluorescent PAH molecules are also common constituents of other complex
particles, including soot particle agglomerates. It has been observed that the fluorescent emission
of PAH constituents on soot particles can be weak due to quenching from the bulk material
(Panne et al., 2000). Several examples of soot particles shown in Figure 2g are fluorescent in
FL1 and indeed should be considered as interfering particle types, as will be discussed. Three
miscellaneous particles (laboratory wipes and two colors of cotton t-shirts) were also
interrogated by rubbing samples over the WIBS inlet, because of their relevance to indoor
aerosol investigation (e.g. Bhangar et al., 2014;Handorean et al., 2015;e.g. Bhangar et al., 2016).
These particles (dark blue dots, Fig. 2 bottom row) show varying median intensity in FL1,
suggesting that sources such as tissues, cleaning wipes, and cotton clothing could be sources of
fluorescent particles within certain built environments.

Another interesting point from the observations of median fluorescence intensity is that the
three viable bacteria aerosolized in this study each shows moderately fluorescent characteristics
in FL1 and low fluorescent characteristics in FL2 and FL3 (Fig. 2a-c). A study by Hernandez et
al. (2016) also focused on analysis strategies using the WIBS and shows similar results regarding
bacteria. Of the 14 bacteria samples observed in the Hernandez et al. study, 13 were categorized
as predominantly A-type particles, thus meaning they exhibited fluorescent properties in FL1 and
only a very small fraction of particles showed fluorescence above the applied threshold (FT +
3σ) in either FL2 or FL3. The FL3 channel in the WIBS-4A has an excitation of 370 nm and
emission band of 420-650 nm, similar to that of the UV-APS with an excitation of 355 nm and
emission band of 420-575 nm. Previous studies have suggested that viable microorganisms (i.e.
bacteria) show fluorescence characteristics in the UV-APS due to the excitation source of 355
nm that was originally designed to excite NAD(P)H and riboflavin molecules present in actively
metabolizing organisms (Agranovski et al., 2004;Hairston et al., 1997;Ho et al., 1999;Pöhlker et
al., 2012). Previous studies with the UV-APS and other UV-LIF instruments using
approximately similar excitation wavelengths have shown a strong sensitivity to the detection of
"viable" bacteria (Hill et al., 1999b;Pan et al., 1999;Hairston et al., 1997;Brosseau et al., 2000).
Because the bacteria here were aerosolized and detected immediately after washing from growth
media, we expect that a high fraction of the bacterial signal was a result of living vegetative
bacterial cells. The results presented here and from other studies using WIBS instruments, in
contrast to reports using other UV-LIF instruments, suggest that the WIBS-4A is highly sensitive
to the detection of bacteria using 280 nm excitation (only FL1 emission), but less so using the
370 nm excitation (FL3 emission) (e.g. Perring et al., 2015;Hernandez et al., 2016). A study by
Agranovski et al. (2003) also demonstrated that the UV-APS was limited in its ability to detect
endospores (reproductive bacterial cells from spore-forming species with little or no metabolic
activity and thus low NAD(P)H concentration).  The lack of FL3 emission observed from
bacteria in the WIBS may also suggest a weaker excitation intensity in Xe2 with respect to Xe1,



manifesting in lower overall FL3 emission intensity (Könemann et al., In Prep.). Gain voltages
applied differently to PMT2 and PMT3 could also impact differences in relative intensity
observed. Lastly, it has been proposed that the rapid sequence of Xe1 and Xe2 excitation could
lead to quenching of fluorescence from the first excitation flash, leading to overall reduced
fluorescence in the FL3 channel (Sivaprakasam et al., 2011). These factors may similarly affect
all WIBS instruments and should be kept in mind when comparing results here with other UV-
LIF instrument types.

**4.2 Fluorescence type varies with particle size**

The purpose of Figure 2 is to distill complex distributions of the five data parameters into a
single value for each in order to show broad trends that differentiate biological and non-
biological particles. By representing the complex data in such a simple way, however, many
relationships are averaged away and lost. For example, the histogram of FL1 intensity for fungal
spore *Aspergillus niger* (Fig. S3) shows a broad distribution with long tail at high fluorescence
intensity, including ca. ~ 6 % of particles that saturate the FL1 detector (Table S2). If a given
distribution were perfectly Gaussian and symmetric, the mean and standard deviation values
would be sufficient to fully describe the distribution. However, given that asymmetric
distributions often include detector-saturating particles, no single statistical fit characterizes data
for all particle types well. Median values were chosen for Figure 2 knowing that the resultant
values can reduce the physical meaning in some cases. For example, the same *Aspergillus niger*
particles show a broad FL1 peak at ~150 a.u. and another peak at 2047 a.u. (detector saturated),
whereas the median FL1 intensity is 543 a.u., at which point there is no specific peak.  In this
way, the median value only broadly represents the data by weighting both the broad distribution
and saturating peak. To complement the median values, however, Table 1 also shows the fraction
of particles that were observed to saturate the fluorescence detector in each channel.
The representation of median values for each of the five parameters (Fig. 2) shows broad
separation between particle classes, but discriminating more finely between particle types with
similar properties by this analysis method can be practically challenging. Rather than
investigating the intensity of fluorescence emission in each channel, however, a common method
of analyzing field data is to apply binary categorization for each particle in each fluorescence
channel. For example, by this process, a particle is either fluorescent in a given FL channel
(above emission intensity threshold) or non-fluorescent (below threshold). In this way, many of
the challenges of separation introduced above are significantly reduced, though others are
introduced. Perring et al. (2015) introduced a WIBS classification strategy by organizing
particles sampled by the WIBS as either non-fluorescent or into one of seven fluorescence types
(e.g. Fig. 1).
Complementing the perspective from Figure 2, stacked particle type plots (Fig. 3) show
qualitative differences in fluorescence emission by representing different fluorescence types as
different colors. The most important observation here is that almost all individual biological
particles aerosolized (top two rows of Fig. 3) are fluorescent, meaning that they exhibit
fluorescence emission intensity above the standard threshold (FT baseline + 3σ) in at least one
fluorescence channel and are depicted with a non-gray color. Figure S4 shows the stacked
particle type plots for all 69 materials analyzed in this study as a comprehensive library. In
contrast to the biological particles, most particles from non-biological origin were observed not



to show fluorescence emission above the threshold in any of the fluorescence channels and are
thus colored gray. For example, 11 of the 15 samples of dust aerosolized show <15% of particles
to be fluorescent at particle sizes <4 µm.  Similarly, 4 of 5 samples of HULIS aerosolized show
<7 % of particles to be fluorescent at particle sizes <4 µm. The size cut-point here was chosen
arbitrarily to summarize the distributions. Two examples shown in Figure 3 (Dust 10 and HULIS
3) are representative of average dust and HULIS types analyzed, respectively, and are relatively
non-fluorescent. Of the four dust types that exhibit a higher fraction of fluorescence, two (Dust 3
and Dust 4) are relatively similar and show ~75% fluorescent particles <4 µm, with particle type
divided nearly equally across the A, B, and AB particle types (Fig. S4I). The two others (Dust 2
and Dust 6) show very few similarities between one another, where Dust 2 shows size-dependent
fluorescence and Dust 6 shows particle type A and B at all particle sizes (Fig. S4I). As seen by
the median fluorescence intensity representation (Fig. 2, Table 1), however, the relative intensity
in each channel for all dusts is either below or only marginally above the fluorescence threshold.
Thus, the threshold value becomes critically important and can dramatically impact the
classification process, as will be discussed in a following section. Similarly, HULIS 5 (Fig. S4K)
is the one HULIS type that shows an anomalously high fraction of fluorescence, and is
represented by B, C, BC particle types, but at intensity only marginally above the threshold value
and at 0% detector saturation in each channel.

Several types of non-biological particles, specifically brown carbon and combustion soot and
smoke, exhibited higher relative fractions of fluorescent particles compared to other non-
biological particles. Two of the three types of brown carbon sampled show >50% of particles to
be fluorescent at sizes >4 µm (Figs. 3i, l), though their median fluorescence is relatively low and
neither shows saturation in any of the three fluorescent channels. Out of six soot samples
analyzed, four showed >69% of particles to be fluorescent at sizes >4 µm, most of which are
dominated by B particle types. Two samples of combustion soot are notably more highly
fluorescent, both in fraction and intensity. Soot 3 (fullerene soot) and Soot 4 (diesel soot) show
FL1 intensity of 318 a.u. and 751 a.u., respectively, and are almost completely represented as A
particle type. The fullerene soot is not likely a good representative of most atmospherically
relevant soot types, however diesel soot is ubiquitous in anthropogenically-influenced areas
around the world. The fact that it exhibits high median fluorescence intensity implies that
increasing the baseline threshold slightly will not appreciably reduce the fraction of particles
categorized as fluorescent, and these particles will thus be counted as fluorescent in many
instances. The one type of wood smoke analyzed (Soot 6) shows ca. 70% fluorescent at >4 µm,
mostly in the B category, with moderate to low FL2 signal, and also presents similarly as
cigarette smoke. Additionally, the two smoke samples in this study (Soot 5, cigarette smoke and
Soot 6, wood smoke) share similar fluorescent particle type features with two of the brown
carbon samples BrC 1 and BrC2. The smoke samples are categorized predominantly as B-type
particles, whereas samples more purely comprised of soot exhibit predominantly A-type
fluorescence. This distinction between smoke and soot may arise partially because the smoke
particles are complex mixtures of amorphous soot with condensed organic liquids, indicating that
compounds similar to the brown carbon analyzed here could heavily influence the smoke particle
signal.

Biological particle types were chosen for Figure 3 to show the most important trends among
all particle types analyzed. Two pollen are shown here to highlight two common types of
fluorescence properties observed. Pollen 9 (Fig. 3a) shows particle type transitioning between A,



AB, and ABC as particle size gets larger. Pollen 9 (*Phleum pretense*) has a physical diameter of
~35 µm, so the mode seen in Figure 3a may be a result of fragmented pollen and due to the upper
particle size limit of WIBS detection, intact pollen cannot be detected (Pöhlker et al., 2013).
Pollen 8 (Fig. 3d) shows a mode peaking at ~10 µm in diameter and comprised of a mixture of
B, AB, BC, and ABC particles as well as a larger particle mode comprised of ABC particles. The
large particle mode appears almost monodisperse, but this is due to the WIBS ability to sample
only the tail of the distribution due to the upper size limit of particle collection (~20 µm as
operated). It is important to note that excitation pulses from the Xe flash lamps are not likely to
penetrate the entirety of large pollen particles, and so emission information is likely limited to
outer layers of each pollen grain. Excitation pulses can penetrate a relatively larger fraction of
the smaller pollen fragments, however, meaning that the differences in observed fluorescence
may arise from differences the layers of material interrogated. Fungi 1 (Fig. 3b) was chosen
because it depicts the most commonly observed fluorescence pattern among the fungal spore
types analyzed (~3 µm mode mixed with A and AB particles). Fungi 4 (Fig. 3e) represents a
second common pattern (particle size peaking at larger diameter, minimal A-type, and dominated
by AB, ABC particle types). All three bacteria types analyzed were dominated by A-type
fluorescence. One gram-positive (Bacteria 1) and one gram-negative bacteria (Bacteria 3) types
are shown in Figure 3c, f, respectively.

**4.3 Fluorescence intensity varies strongly with particle size**

An extension of observation from the many particle classes analyzed is that particle type (A,
AB, ABC, etc.) varies strongly as a function of particle size. This is not surprising, given that it
has been frequently observed and reported that particle size significantly impacts fluorescence
emission intensity (e.g. Hill et al., 2001;Sivaprakasam et al., 2011). The higher the fluorescent
quantum yield of a given fluorophore, the more likely it is to fluoresce. For example, pure
biofluorophores (middle row of Fig. 2) and PAHs (bottom row of Fig. 2) have high quantum
yields and thus exhibit relatively intense fluorescence emission, even for particles <1 µm. In
contrast, more complex particles comprised of a wide mixture of molecular components are
typically less fluorescent per volume of material.  At small sizes the relative fraction of these
particles that fluoresce is small, but as particles increase in size they are more likely to contain
enough fluorophores to emit a sufficient number of photons to record an integrated light intensity
signal above a given fluorescence threshold. Thus, the observed fluorescence intensity scales
approximately between the 2$^{nd}$ and 3$^{rd}$ power of the particle diameter (Sivaprakasam et al.,
2011;Taketani et al., 2013;Hill et al., 2015).

The general trend of fluorescence dependence on size is less pronounced for FL1 than for
FL2 and FL3. This can be seen by the fact that the scatter of points along the FL1 axis in Figure
2b is not clearly size-dependent and is strongly influenced by particle type (i.e. composition
dependent). In Figure 2c, however, the median points cluster near the vertical (size) axis and
both FL2 and FL3 values increase as particle size increases. It is important to note, however, that
the method chosen for particle generation in the laboratory strongly impacts the size distribution
of aerosolized particles. For example, higher concentrations of an aqueous suspension of particle
material generally produce larger particles, and the mechanical force used to agitate powders or
aerosolize bacteria can have strong influences on particle viability and physical agglomeration or
fragmentation of the aerosol (Mainelis et al., 2005). So, while the absolute size of particles
shown here is not a key message, the relative fluorescence at a given size can be informative.



As discussed, each individual particle shows increased probability of exhibiting fluorescence emission above a given fluorescence threshold as size increases. Using Pollen 9 (*Phleum pratense*, Fig. 3a) as an example, most particles <3 µm show fluorescence in only the FL1 channel and are thus classified as A-type particles. For the same pollen, however, particles ca. 2-6 µm in diameter are more likely to be recorded as AB-type particles, indicating that they have retained sufficient FL1 intensity, but have exceeded the FL2 threshold to add B-type fluorescence character. Particles larger still (>4 µm) are increasingly likely to exhibit ABC character, meaning that the emission intensity in the FL3 channel has increased to cross the fluorescence threshold. Thus, for a given particle type and a constant threshold as a function of particle size, the relative breakdown of fluorescence type changes significantly as particle size increases. The same general trend can be seen in many other particle types, for example Pollen 8 (*Alnus glutinosa*, Fig. 3d), Fungi 1 (*Aspergillus brasiliensis*, Fig. 3b), and to a lesser degree HULIS 3 (Suwannee fulvic acid, Fig. 3j) and Brown Carbon 2 (Fig. 3i). The "pathway" of change, for Pollen 9, starts as A-type at small particle size and adds B and eventually ABC (A→AB→ABC), whereas Pollen 8 starts primarily with B-type at small particle size and separately adds either B or C en route to ABC (B→AB or BC→ABC). In this way, not only is the breakdown of fluorescence type useful in discriminating particle distributions, but the pathway of fluorescence change with particle size can also be instructive.

To further highlight the relationship between particle size and fluorescence, four kinds of particles (Dust 2, HULIS 5, Fungi 4, and Pollen 9) were each binned into 4 different size ranges, and the relative number fraction was plotted versus fluorescence intensity signal for each channel (Fig. 4). In each case, the fluorescence intensity distribution shifts to the right (increases) as the particle size bin increases. This trend is strongest in the FL2 and FL3 (middle and right columns of Fig. 4) for most particle types, as discussed above.

The fact that particle fluorescence type can change so dramatically with increasing particle size becomes critically important when the Perring-style particle type classification is utilized for laboratory or field investigation. For example Hernandez et al. (2016) aerosolized a variety of species of pollen, fungal spores, and bacteria in the laboratory and presented the break-down of particle types for each aerosolized species. This first comprehensive overview summarized how different types of biological material (i.e. pollen and bacteria) might be separated based on their fluorescence properties when presented with a population of relatively monodisperse particles. This was an important first step, however, differentiation becomes more challenging when broad size distributions of particles are mixed in an unknown environment. In such a case, understanding how the particle type may change as a function of particle size may become an important aspect of analysis.

**4.4 Fluorescence threshold defines particle type**

Particle type analysis is not only critically affected by size, but also by the threshold definition chosen. Figure 5 represents the same matrix of particle types as in Figure 3, but shows the fluorescence intensity distribution in each channel (at a given narrow range of sizes in order to minimize the sizing effect on fluorescence). Figure 5 can help explain the breakdown of particle type (and associated colors) shown in Figure 3. For example, in Figure 5a, the median fluorescence intensity in FL1 for Pollen 9 (2046 a.u., detector saturated) in the size range 3.5-4.0 µm far exceeds the 3σ threshold (51 a.u.), and so essentially all particles exhibit FL1 character.





Approximately 90% of particles of Pollen 9 are above the 3σ FL2 threshold (25 a.u.), and
approximately 63% of particles are above the 3σ FL3 threshold (49 a.u). These three channels of
information together describe the distribution of particle type at the same range of sizes:  9% A,
26% AB, 63% ABC, and 2% other categories. Since essentially all particles are above the
threshold for FL1, particles are thus assigned as A type particles (if < FL2 and FL3 thresholds),
AB (if >FL2 threshold and <FL3 threshold), or ABC (if > FL2 and FL3 thresholds). Thus, the
distribution of particles at each fluorescence intensity and in relation to a given thresholding
strategy defines the fluorescence type breakdown and the pathway of fluorescence change with
particle size. It is important to note differences in this pathway for biofluorophores (Figs. S4G
and S4H). For example Biofluorophore 1 (riboflavin) follows the pathway B→C→BC while
Biofluorophore 11 (tryptophan) follows the pathway A→BC→ABC.
By extension, the choice of threshold bears heavily on how a given particle breakdown
appears and thus how a given instrument may be used to discriminate between biological and
non-biological particles. A commonly made assumption is that particles exhibiting fluorescence
by the WIBS (or UV-APS) can be used as a lower limit proxy to the concentration of biological
particles, though it is known that interfering particle types confound this simple assumption
(Huffman et al., 2010). Increasing the fluorescence threshold can reduce categorizing weakly
fluorescent particles as biological, but can also remove weakly fluorescing biological particles of
interest (Huffman et al., 2012). Figure 6 provides an analysis of 8 representative particle types (3
biological, 5 non-biological) in order to estimate the trade-offs of increasing fluorescence
threshold separately in each channel. Once again, the examples chosen here represent general
trends and outliers, as discussed previously for Figure 3. Four threshold strategies are presented:
three as the instrument fluorescence baseline plus increasing uncertainty on that signal (FT + 3σ,
FT + 6σ, and FT + 9σ), as well as the FP3 strategy suggested by Wright et al. (2014). Using Dust
4 as an example (Fig. 6d), by increasing the threshold from 3σ (red traces) to 6σ (orange traces),
the fraction of dust particles fluorescent in FL1 decreases from approximately 50% to 10%.
Increasing the fluorescence threshold even higher to 9σ, reduces the fraction of fluorescence to
approximately 1%, thus eliminating nearly all interfering particles of Dust 3. In contrast, for
biological particles such as Pollen 9 (Fig. 6b), increasing the threshold from 3σ to 9σ does very
little to impact the relative breakdown of fluorescence category or the fraction of particles
considered fluorescent in at least one channel. Changing threshold from 3σ to 9σ decreases the
FL1 fraction minimally (98.3% to 97.9%), and for FL2 and FL3 the fluorescence fraction
decreases from 90% to 50% and from 60% to 42%, respectively. Figure 6 also underscores how
increasing particle size affects fluorescence fraction, as several particle types (e.g. Pollen 9 and
HULIS 5) show sigmoidal curves that proceed toward the right (lower fraction at a given size) as
the threshold applied increases and thus removes more weakly fluorescent particles.
To better understand how the different thresholding strategies qualitatively change the
distribution of particle fluorescence type, Figure 7 shows stacked fluorescence type distributions
for each of the four thresholds analyzed. Looking first at Dust 3 (Fig. 7d), the standard threshold
definition of 3σ shows approximately 80% of particles to be fluorescent in at least one channel,
resulting in a distribution of predominantly A, B, and AB-type particles. As the threshold is
increased, however, the total percentage of fluorescent particles decreases dramatically to 1% at
9σ and the particle type of the few remaining particles shifts to A-type particles. A similar trend
of fluorescent fraction can also be seen for Soot 6 (wood smoke) and Brown Carbon 2, where
almost no particle (10% and 16%, respectively) remain fluorescent using the 9σ threshold. Soot 4



(diesel soot), in contrast, exhibits the same fraction and breakdown of fluorescent particles
whether using the 3σ or 9σ threshold. Using the FP3 threshold (which employs very high FL1
threshold), however, the fluorescent properties of the diesel soot change dramatically to non-
fluorescent. As a 'worst case' scenario, HULIS 5 shows ca. 60% of particles to be fluorescent
using the 3σ threshold. In this case, increasing the threshold from 6σ to 9σ only marginally
decreases the fraction of fluorescent particles to ca. 35% and 22%, respectively, and the break-
down remains relatively constant in B, C, and BC types. Changing the threshold definition to
FP3 in this case also does not significantly change the particle type break-down, since the high
FP3 threshold applies only to FL1.
As stated, the WIBS is mostly often applied toward the detection and characterization of
biological aerosol particles. For the biological particles analyzed (Fig. 7, top rows), increasing
the  threshold  from 3σ to 9σ shows only a marginal decrease in the total fluorescent fraction for
Pollen 9, Fungal Spore 1, and Bacteria 1, and only a slight shift in fluorescence type as a
function of size. Using the FP3 threshold, however, for each of the three biological species the
non-fluorescent fraction increases substantially. Wright et al. (2014) found that the FP3 threshold
definition showed a strong correlation with ice nucleating particles and the authors suggested
these particles with high FL1 intensity were likely to be fungal spores. This may have been the
case, but given the analysis here, the FP3 threshold is also likely to significantly underestimate
fungal spore number by missing weakly or marginally fluorescent spores.
Based on the threshold analysis results shown in Figure 7, marginally increasing the
threshold in each case may help eliminate non-biological, interfering particles without
significantly impacting the number of biological particles considered fluorescent. Each threshold
strategy brings trade-offs, and individual users must understand these factors to make appropriate
decisions for a given scenario. These data suggest that using a threshold definition of FT baseline
+ 9σ is likely to reduce interferences from most non-biological particles without significantly
impacting most biological particles.
**4.5 Particle asymmetry varies with particle size**
As a part of the comprehensive WIBS study, particle asymmetry (AF) was analyzed as a
function of particle size for all particles.  As described in Section 2.1, AF in the WIBS-4A is
determined by comparing the symmetry of the forward elastic scattering response of each
particle, measured at the quadrant PMT. Many factors are related to the accuracy of the
asymmetry parameter, including the spatial alignment of the collection optics, signal-to-noise
and dynamic range of the detector, agglomeration of particles with different refractive indices,
and the angle at which a non-symmetrical particle hits the laser (Kaye et al., 2007;Gabey et al.,
2010). Figure 8 shows a summary of the relationship between AF and particle size for all
material types analyzed in Table 1. Soot particles are known to frequently cluster into chains or
rings depending on the number of carbon atoms (Von Helden et al., 1993) and, as a result, can
have long aspect ratios that would be expected to manifest as large AF values. The bacteria
species chosen have rod-like shape features and thus would also exhibit large AF values. These
properties were observed by the WIBS, as two types of soot (diesel and fullerene) and all three
bacteria showed higher AF values than other particles at approximately the same particle
diameter. For an unknown reason, all three brown carbon samples also showed relatively high
AF values given that the individual particles of liquid organic aerosol would be expected to be





spherical with low AF. Similarly, the intact pollen showed anomalously low AF, because a
substantial fraction of each was shown to saturate the WIBS sizing detector, even if the median
particle size (shown) is lower than the saturating value. For this reason we postulate that the side-
scattering detector may not be able to reliably estimate either particle size or AF when particles
are near the sizing limits. Intact pollen, soot samples (diesel and fullerene soot), bacteria and
brown carbon samples were excluded from the linear regression fit, because they appeared
visually as outliers to the trend. All remaining particle groups of material types (7 in total) are
represented by blue in Figure 8. A linear regression $R^2$ value of 0.87 indicates a high degree of
correlation between particle AF and size across the remaining particles. The strong correlation
between these two factors across a wide range of particle types, mixed with the confounding
anomaly of brown carbon, raises a question about the degree to which the asymmetry factor
parameter from the WIBS-4A can be useful or, conversely, to what degree the uncertainty in AF
is dominated by instrumental factors, including those listed above.

## 5. Summary and Conclusions

UV-LIF instruments, including the WIBS, are common tools for the detection and
characterization of biological aerosol particles. The number of commercially available
instruments regularly deployed for ambient monitoring of environmental particle properties is
rising steeply, yet critical laboratory work has been needed to better understand how the
instruments categorize a variety of both biological and non-biological particles. In particular, the
differentiation between weakly fluorescent, interfering particles of non-biological origin and
weakly fluorescing biological particles is very challenging. Here we have aerosolized a
representative list of pollen, fungal spores, and bacteria along with key aerosol types from the
groups of fluorescing non-biological materials expected to be most problematic for UV-LIF
instrumentation.
By analyzing the five WIBS data parameter outputs for each interrogated particle, we have
summarized trends within each class of particles and demonstrated the ability of the instrument
to broadly differentiate populations of particles. The trend of particle fluorescence intensity and
changing particle fluorescence type as a function of particle size was shown in detail. This is
critically important for WIBS and other UV-LIF instrumentation users to keep in mind when
analyzing populations of unknown, ambient particles. In particular, we show that the pathway of
fluorescence particle type change (e.g. A → AB → ABC or B → BC → ABC) with increasing
particle size can be one characteristic feature of unique populations of particles. When
comparing the fluorescence break-down of individual aerosol material types, care should be
taken to limit comparison within a narrow range of particle sizes in order to reduce complexity
due to differing composition or fluorescence intensity effects.
The fluorescence threshold applied toward binary categorization of fluorescence or non-
fluorescent in each channel is absolutely critical to the conceptual strategy that a given user
applies to ambient particle analysis. A standard WIBS threshold definition of instrument
background (FT baseline) + 3σ is commonly applied to discriminate between particles with or
without fluorescence. As has been shown previously, however, any single threshold confounds
simple discrimination of biological and non-biological particles by mixing poorly fluorescent
biological material into non-fluorescent categories, and highly fluorescent non-biological
material into fluorescent categories. Previously introduced thresholding strategies were also used



for comparison. The Wright et al. (2014) definition was shown to aid in removing non-biological particles such as soot, but that it can also lead to the dramatic underestimation of the biological fraction. The strategy utilized by Toprak and Schnaiter (2013) was to define fluorescent biological particles as those with fluorescent characteristics in FL1 and FL3, ignoring any particles with fluorescence in FL2. They proposed this because FL1 shows excitation and emission characteristics well suited for the detection of tryptophan, and FL3 for the detection of NAD(P)H and riboflavin. However, the study here, along with studies by Hernandez et al. (2016) and Perring et al. (2015), have shown that FL2 fluorescence characteristics (B, AB, BC, and ABC type) are common for many types of biological particles and so removing particles with FL2 fluorescence is likely to remove many bioparticles from characterization.

Any one threshold has associated trade-offs and is likely to create some fraction of both false positive and false negative signals. Here we have shown a systematic analysis of four different fluorescence thresholding strategies, concluding that by raising the threshold to FT + 9σ, the reduction in biological material counted as fluorescent is likely to be only minimally effected, while the fraction of interfering material is likely to be reduced almost to zero for most particle types. Several materials exhibiting outlier behavior (e.g. HULIS 5, diesel soot) could present as false positive counts using almost any characterization scheme. It is important to note that HULIS 5 was one of a large number of analyzed particle types and in the minority of HULIS types, however, and it is unclear how likely these highly fluorescent materials are to occur in any given ambient air mass. More studies may be required to sample dusts, HULIS types, soot and smoke, brown organic carbon materials, and various coatings in different real-world settings to better understand how specific aerosol types may contribute to UV-LIF interpretation at a given study location. We also included a comprehensive supplemental document including size distributions for all 69 aerosol materials, stacked by fluorescent particle type and comparing the FT + 3σ and FT + 9σ threshold strategies. These figures are included as a qualitative reference for other instrument users when comparing against laboratory-generated particles or for use in ambient particle interpretation.

It should be noted, however, that the presented assessment is not intended to be exhaustive, but has the potential to guide users of commercial UV-LIF instrumentation through a variety of analysis strategies toward the goal of better detecting and characterizing biological particles. One important note is that the information presented here is strongly instrument dependent due to fluorescence PMT voltages and gains, specific fluorescence calibrations applied, and other instrument parameters (Robinson et al., 2017). For example, the suggested particle type classification introduced by Perring et al. (2015), will vary somewhat between instruments, though more work will be necessary to determine the magnitude of these changes. Thus, we do not introduce these data primarily as a library to which all other WIBS instrument should be compared rigorously, but rather as general trends that are expected to hold broadly true.

Several examples of strongly fluorescing particles of specific importance to the built environment (e.g. cellulose fibers, particles from cotton t-shirts, and laboratory wipes) show that these particle types could be very important sources of fluorescent particles indoors (i.e. Figs. S4S and S4T). This will also require further study, but should be taken seriously by researchers who utilize UV-LIF instrumentation to estimate concentrations and properties of biological material within homes, indoor occupational environments, or hospitals.





The study presented here is meant broadly to achieve two aims. The first aim is to present a summary of fluorescent properties of the most important particle types expected in a given sample and to suggest thresholding strategies (i.e. FT + 9σ) that may be widely useful for improving analysis quality. The second aim is to suggest key analysis and plotting strategies that other UV-LIF, especially WIBS, instrumentation users can utilize to interrogate particles using their own instruments. By proposing several analysis strategies we aim to introduce concepts to the broader atmospheric community in order to promote deeper discussions about how best to continue improving UV-LIF instrumentation and analyses.

## 6. Acknowledgments

The authors acknowledge the University of Denver for financial support from the faculty start-up fund. Nicole Savage acknowledges financial support from the Phillipson Graduate Fellowship at the University of Denver. Christine Krentz acknowledges financial support from the Summer Undergraduate Research Grant program through the Undergraduate Research Center at the University of Denver. Tobias Könemann and Christopher Pöhlker acknowledge financial support by the Max Planck Society and the Max Planck Graduate Center with the Johannes Gutenberg-Universität Mainz (MPGC). Gediminas Mainelis acknowledges support by the New Jersey Agricultural Experiment Station (NJAES) at Rutgers, The State University of New Jersey. Ulrich Pöschl and Meinrat O. Andreae are acknowledged for useful discussions and support for the authors. Gavin McMeeking from Handix Scientific is acknowledged for the development of the WIBS analysis toolkit. Martin Gallagher, Jonathan Crosier, and the Department of Geology and Earth Science in the School of Earth and Environmental Sciences, University of Manchester provided several samples of raw materials. Marie Gosselin is acknowledged for discussions about WIBS analysis, and Ben Swanson is acknowledged for help with the conceptual design of figures.



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

Table 1. Fluorescence values of standard PSLs, determined as the peak (mean) of a Gaussian fit
applied to a histogram of the fluorescence signal in each channel. Uncertainties are one standard
deviation from the Gaussian mean.

|  | FL1 | FL2 | FL3 |
|---|---|---|---|
| 2 µm Green | 69 ± 49 | 1115 ± 57 | 214 ± 29 |
| 2 µm Red | 44 ± 30 | 160 ± 18 | 28 ± 13 |
| 2.1 µm Blue | 724± 111 | 1904 ± 123 | 2045 ± 6 |






Table 2. Median values for each of the five data parameters, along with percent of particles that
saturate fluorescence detector in each fluorescence channel. Uncertainty (as one standard
deviation, σ) listed for particle size and asymmetry factor (AF). Only a sub-selection of pollen
are characterized as fragmented pollen because not all pollen presented the smaller size fraction
or fluorescence characteristics that represent fragments.

| Materials | | FL1 | FL1 Sat % | FL2 | FL2 Sat % | FL3 | FL3 Sat % | Size (µm) | AF | Aerosolization method |
|---|---|---|---|---|---|---|---|---|---|---|
| BIOLOGICAL MATERIALS | | | | | | | | | | |
| Pollen | | | | | | | | | | |
| Intact Pollen | | | | | | | | | | |
| 1 | *Urtica diocia (Stinging Nettle)* | 2047.0 | 99.2 | 2047.0 | 99.4 | 1072.0 | 9.9 | 16.9 ± 2.2 | 18.5 ± 8.3 | Powder (P1) |
| 2 | *Artemisia vulgaris (Common Mugwort)* | 1980.0 | 48.3 | 2047.0 | 99.7 | 2047.0 | 90.3 | 19.7 ± 1.0 | 14.2 ± 7.6 | Powder (P1) |
| 3 | *Castanea sativa (European Chestnut)* | 830.0 | 19.3 | 258.0 | 2.9 | 269.0 | 0.8 | 15.3 ± 1.7 | 17.0 ± 9.5 | Powder (P1) |
| 4 | *Corylus avellana (Hazel)* | 1371.0 | 44.4 | 532.0 | 5.6 | 99.0 | 2.8 | 16.6 ± 2.1 | 24.2 ± 12.6 | Powder (P1) |
| 5 | *Taxus baccata (Common Yew)* | 525.0 | 0.4 | 561.0 | 0.2 | 615.0 | 0.0 | 16.0 ± 1.3 | 22.2 ± 10.0 | Powder (P1) |
| 6 | *Rumex acetosella (Sheep Sorrel)* | 2047.0 | 73.5 | 2047.0 | 55.1 | 693.0 | 2.7 | 16.2 ± 2.0 | 21.7 ± 10.8 | Powder (P1) |
| 7 | *Olea europaea (European Olive Tree)* | 131.0 | 1.1 | 395.0 | 0.4 | 119.0 | 0.0 | 19.7 ± 1.2 | 17.7 ± 7.6 | Powder (P1) |
| 8 | *Alnus glutinosa (Black Alder)* | 109.0 | 3.3 | 432.0 | 1.2 | 102.0 | 0.9 | 18.6 ± 1.7 | 15.8 ± 8.5 | Powder (P1) |
| 9 | *Phleum pratense (Timothy Grass)* | 2047.0 | 100.0 | 2012.0 | 49.8 | 651.0 | 1.9 | 15.1 ± 1.7 | 24.1 ± 12.2 | Powder (P1) |
| 10 | *Populus alba (White Poplar)* | 2047.0 | 95.9 | 2047.0 | 92.2 | 1723.0 | 39.2 | 18.7 ± 1.9 | 21.2 ± 10.4 | Powder (P1) |
| 11 | *Taraxacum officinale (Common Dandelion)* | 2047.0 | 99.1 | 1309.0 | 21.8 | 1767.0 | 44.2 | 15.4 ± 1.8 | 22.2 ± 11.9 | Powder (P1) |
| 12 | *Amaranthus retroflexus (Redroot Amaranth)* | 980.0 | 36.7 | 1553.0 | 36.7 | 1061.0 | 18.0 | 17.7 ± 2.2 | 19.4 ± 12.1 | Powder (P1) |
| 13 | *Aesculus hippocastanum (Horse-chestnut)* | 762.0 | 23.5 | 876.0 | 23.5 | 776.0 | 23.5 | 16.2 ± 2.0 | 22.2 ± 13.4 | Powder (P1) |
| 14 | *Lycopodium (Clubmoss)* | 40.0 | 0.1 | 32.0 | 0.0 | 27.0 | 0.0 | 3.9 ± 1.86 | 24.5 ± 15.9 | Powder (P1) |
| | | | | | | | | | | |
| Fragment Pollen | | | | | | | | | | |
| 3 | *Castanea sativa (European Chestnut)* | 74.0 | 11.0 | 113.0 | 0.4 | 84.0 | 0.1 | 7.0 ± 3.1 | 24.6 ± 13.7 | Powder (P1) |
| 4 | *Corylus avellana (Hazel)* | 263.0 | 28.8 | 119.0 | 0.5 | 46.0 | 0.2 | 6.1 ± 3.7 | 20.4 ± 13.7 | Powder (P1) |
| 5 | *Taxus baccata (Common Yew)* | 40.0 | 0.2 | 28.0 | 0.1 | 34.0 | 0.0 | 2.6 ± 2.2 | 16.0 ± 12.2 | Powder (P1) |
| 6 | *Rumex acetosella (Sheep Sorrel)* | 417.0 | 87.1 | 88.0 | 0.4 | 71.0 | 0.1 | 6.0 ± 2.5 | 24.4 ± 12.4 | Powder (P1) |
| 7 | *Olea europaea (European Olive Tree)* | 40.0 | 1.9 | 22.0 | 0.1 | 33.0 | 0.0 | 2.6 ± 1.6 | 10.4 ± 9.3 | Powder (P1) |
| 8 | *Alnus glutinosa (Black Alder)* | 46.0 | 4.6 | 46.0 | 0.3 | 44.0 | 0.2 | 6.1 ± 3.2 | 25.2 ± 14.6 | Powder (P1) |
| 9 | *Phleum pretense (Timothy Grass)* | 2047.0 | 85.5 | 129.0 | 1.2 | 63.0 | 0.1 | 6.0 ± 3.2 | 23.1 ± 13.4 | Powder (P1) |
| 10 | *Populus alba (White Poplar)* | 642.0 | 35.2 | 237.0 | 8.6 | 103.0 | 0.5 | 7.4 ± 4.0 | 24.7 ± 14.2 | Powder (P1) |
| 11 | *Taraxacum officinale (Common Dandelion)* | 2047.0 | 71.9 | 195.0 | 0.4 | 88.0 | 0.8 | 6.1 ± 3.1 | 23.7 ± 13.5 | Powder (P1) |



| | | | | | | | | | | |
|---|---|---|---|---|---|---|---|---|---|---|
| 12 | *Amaranthus retroflexus (Redroot Amaranth)* | 104.0 | 15.6 | 138.0 | 5.6 | 101.0 | 3.4 | 7.3 ± 2.8 | 27.7 ± 14.6 | Powder (P1) |
| 13 | *Aesculus hippocastanum (Horse-chestnut)* | 43.0 | 6.0 | 106.0 | 0.2 | 42.0 | 0.2 | 4.3 ± 3.1 | 19.7 ± 13.4 | Powder (P1) |
| | | | | | | | | | | |
| **Fungal spores** | | | | | | | | | | |
| 1 | *Aspergillus brasiliensis* | 1279.0 | 38.5 | 22.0 | 0.0 | 33.0 | 0.0 | 3.6 ± 1.8 | 20.8 ± 10.3 | Fungal |
| 2 | *Aspergillus niger; WB 326* | 543.0 | 6.2 | 18.0 | 0.0 | 29.0 | 0.0 | 2.7 ± 0.9 | 17.1 ± 10.7 | Fungal |
| 3 | *Rhizopus stolonifera (Black Bread Mold); UNB-1* | 78.0 | 11.2 | 20.0 | 0.1 | 34.0 | 0.1 | 4.4 ± 2.3 | 21.4 ± 14.4 | Fungal |
| 4 | *Saccharomyces cerevisiae (Brewer's Yeast)* | 2047.0 | 96.6 | 97.0 | 0.3 | 41.0 | 0.1 | 7.2 ± 3.7 | 28.7 ± 16.8 | Fungal |
| 5 | *Aspergillus versicolor; NRRL 238* | 2047.0 | 78.2 | 55.0 | 0.0 | 40.0 | 0.0 | 4.5 ± 2.5 | 24.5 ± 16.9 | Fungal |
| | | | | | | | | | | |
| **Bacteria** | | | | | | | | | | |
| 1 | *Bacillus atrophaeus* | 443.0 | 1.0 | 10.0 | 0.0 | 36.0 | 0.0 | 2.2 ± 0.4 | 17.4 ± 4.1 | Bacterial |
| 2 | *Escherichia coli* | 454.0 | 1.4 | 12.0 | 0.0 | 33.0 | 0.0 | 1.2 ± 0.3 | 19.3 ± 2.8 | Bacterial |
| 3 | *Pseudomonas Stutzeri* | 675.0 | 0.4 | 16.0 | 0.0 | 36.0 | 0.0 | 1.1 ± 0.3 | 19.2 ± 2.8 | Bacterial |
| | | | | | | | | | | |
| **Biofluorophores** | | | | | | | | | | |
| 1 | Riboflavin | 41.0 | 0.0 | 190.0 | 2.5 | 119.0 | 1.3 | 2.5 ± 2.5 | 13.2 ± 12.2 | Powder (P1) |
| 2 | Chitin | 116.5 | 6.2 | 61.0 | 0.1 | 40.0 | 0.0 | 2.7 ± 2.1 | 16.1 ± 13.5 | Powder (P1) |
| 3 | NAD | 49.0 | 0.2 | 962.0 | 26.7 | 515.0 | 15.0 | 2.1 ± 2.2 | 12.2 ± 10.1 | Powder (P1) |
| 4 | Folic Acid | 41.0 | 0.0 | 34.0 | 0.1 | 28.0 | 0.1 | 3.7 ± 3.4 | 18.6 ± 13.6 | Powder (P1) |
| 5 | Cellulose, fibrous medium | 54.0 | 0.2 | 37.0 | 0.1 | 27.0 | 0.0 | 3.7 ± 2.5 | 20.4 ± 15.7 | Powder (P1) |
| 6 | Ergosterol | 2047.0 | 81.8 | 457.0 | 2.6 | 355.0 | 11.6 | 6.8 ± 4.0 | 22.6 ± 12.9 | Powder (P1) |
| 7 | Pyrdoxine | 661.0 | | 39.0 | | 28.0 | | 1.0 ± 0.2 | 20.0 ± 13.0 | Powder (P1) |
| 8 | Pyridoxamine | 706.0 | 10.7 | 40.0 | 0.0 | 28.0 | 0.0 | 5.2 ± 2.5 | 20.2 ± 12.7 | Powder (P1) |
| 9 | Tyrosine | 2047.0 | 59.7 | 42.0 | 0.0 | 29.0 | 0.0 | 2.9 ± 3.4 | 15.4 ± 11.6 | Powder (P1) |
| 10 | Phenylalanine | 53.0 | 0.0 | 29.0 | 0.0 | 24.0 | 0.0 | 3.2 ± 2.0 | 21.1 ± 15.4 | Powder (P1) |
| 11 | Tryptophan | 2047.0 | 78.0 | 357.0 | 9.0 | 30.0 | 0.0 | 3.5 ± 2.9 | 20.9 ± 17.0 | Powder (P1) |
| 12 | Histidine | 59.0 | 0.2 | 29.0 | 0.0 | 25.0 | 0.0 | 2.0 ± 1.7 | 11.6 ± 10.0 | Powder (P1) |
| | | | | | | | | | | |
| **NON-BIOLOGICAL MATERIALS** | | | | | | | | | | |
| **Dust** | | | | | | | | | | |
| 1 | Arabic Sand | 48.0 | 0.1 | 37.0 | 0.0 | 29.0 | 0.0 | 3.1 ± 2.2 | 16.1 ± 15.7 | Powder (P3) |
| 2 | California Sand | 66.0 | 1.1 | 42.0 | 0.0 | 31.0 | 0.0 | 4.0v1.9 | 18.8 ± 14.6 | Powder (P2) |
| 3 | Africa Sand | 88.0 | 0.0 | 48.0 | 0.0 | 26.0 | 0.0 | 2.2 ± 1.4 | 15.3 ± 11.0 | Powder (P2) |
| 4 | Murkee-Murkee Australian Sand | 88.0 | 0.7 | 47.0 | 0.0 | 26.0 | 0.0 | 1.9 ± 1.1 | 10.9 ± 9.2 | Powder (P2) |




| # | Name | | | | | | | | | Form |
|---|------|---|---|---|---|---|---|---|---|------|
| 5 | Manua Key Summit Hawaii Sand | 54.0 | 0.1 | 33.0 | 0.0 | 25.0 | 0.0 | 1.5 ± 0.7 | 10.8 ± 13.4 | Powder (P2) |
| 6 | Quartz | 66.0 | 0.0 | 38.0 | 0.0 | 24.0 | 0.0 | 1.7 ± 0.8 | 11.2 ± 12.7 | Powder (P2) |
| 7 | Kakadu Dust | 58.0 | 0.0 | 35.0 | 0.0 | 25.0 | 0.0 | 2.7 ± 1.4 | 15.0 ± 12.0 | Powder (P2) |
| 8 | Feldspar | 60.0 | 0.0 | 36.0 | 0.0 | 25.0 | 0.0 | 1.2 ± 0.6 | 10.2 ± 10.6 | Powder (P2) |
| 9 | Hematite | 51.0 | 0.0 | 32.0 | 0.0 | 25.0 | 0.0 | 1.8 ± 1.0 | 10.8 ± 11.9 | Powder (P2) |
| 10 | Gypsum | 49.0 | 0.0 | 30.0 | 0.0 | 26.0 | 0.0 | 4.1 ± 3.0 | 19.3 ± 12.2 | Powder (P2) |
| 11 | Bani AMMA | 48.0 | 0.2 | 31.0 | 0.0 | 26.0 | 0.0 | 3.1 ± 2.1 | 15.8 ± 13.7 | Powder (P2) |
| 12 | Arizona Test Dest | 46.0 | 0.0 | 29.0 | 0.0 | 25.0 | 0.0 | 1.4 ± 0.7 | 10.5 ± 10.5 | Powder (P2) |
| 13 | Kaolinite | 46.0 | 0.0 | 29.0 | 0.0 | 25.0 | 0.0 | 1.5 ± 0.8 | 9.9 ± 10.3 | Powder (P2) |
| | **HULIS** | | | | | | | | | |
| 1 | Waskish Peat Humic Acid Reference | 46.0 | 0.0 | 29.0 | 0.0 | 25.0 | 0.0 | 1.7 ± 0.8 | 10.9 ± 9.8 | Powder (P1) |
| 2 | Suwannee River Humic Acid Standard II | 46.0 | 0.0 | 30.0 | 0.0 | 26.0 | 0.0 | 2.0 ± 1.2 | 13.2 ± 16.5 | Powder (P2) |
| 3 | Suwannee River Fulvic Acid Standard I | 46.0 | 0.0 | 34.0 | 0.0 | 28.0 | 0.0 | 1.7 ± 1.0 | 12.0 ± 10.1 | Powder (P2) |
| 4 | Elliott Soil Humic Acid Standard | 47.0 | 0.0 | 29.0 | 0.0 | 25.0 | 0.0 | 1.2 ± 0.6 | 10.5 ± 10.2 | Powder (P1) |
| 5 | Pony Lake (Antarctica) Fulvic Acid Reference | 46.0 | 0.0 | 49.0 | 0.0 | 37.0 | 0.0 | 2.4 ± 1.8 | 14.0 ± 13.3 | Powder (P2) |
| 6 | Nordic Aquatic Fulvic Acid Reference | 48.0 | 0.1 | 32.0 | 0.0 | 27.0 | 0.0 | 1.8 ± 1.4 | 11.6 ± 9.6 | Powder (P2) |
| | **Polycyclic Hydrocarbons** | | | | | | | | | |
| 1 | Pyrene | 490.0 | 7.4 | 2047.0 | 91.5 | 2047.0 | 81.8 | 5.0 ± 3.5 | 17.4 ± 12.6 | Powder (P1) |
| 2 | Phenanthrene | 2047.0 | 81.9 | 2047.0 | 66.3 | 360.0 | 22.4 | 3.9 ± 3.5 | 14.5 ± 13.6 | Powder (P1) |
| 3 | Naphthalene | 886.0 | 11.6 | 45.0 | 2.1 | 30.0 | 0.7 | 1.1 ± 1.0 | 10.6 ± 9.5 | Powder (P1) |
| | **Combustion Soot and Smoke** | | | | | | | | | |
| 1 | Aquadag | 22.0 | 0.0 | 14.0 | 0.0 | 29.0 | 0.0 | 1.2 ± 0.6 | 10.5 ± 6.6 | Liquid |
| 2 | Ash | 48.0 | 0.2 | 31.0 | 0.0 | 23.0 | 0.0 | 1.7 ± 1.3 | 12.6 ± 11.9 | Powder (P1) |
| 3 | Fullerene Soot | 318.0 | 0.0 | 30.0 | 0.0 | 26.0 | 0.0 | 1.1 ± 0.5 | 17.0 ± 10.6 | Powder (P2) |
| 4 | Diesel Soot | 750.5 | 0.2 | 30.0 | 0.0 | 26.0 | 0.0 | 1.1 ± 0.4 | 21.2 ± 10.1 | Powder (P1) |
| 5 | Cigarette Smoke | 28.0 | 0.6 | 30.0 | 0.1 | 36.0 | 0.0 | 1.0 ± 0.8 | 9.5 ± 4.5 | Smoke |
| 6 | Wood Smoke (*Pinus Nigra ,Black Pine*) | 32.0 | 0.1 | 30.0 | 0.0 | 36.0 | 0.0 | 1.0 ± 0.7 | 9.5 ± 4.3 | Smoke |
| 7 | Fire Ash | 42.0 | 0.2 | 33.0 | 0.0 | 28.0 | 0.0 | 1.8 ± 1.2 | 14.0 ± 16.7 | Powder (P1) |
| | **Brown Carbon** | | | | | | | | | |
| 1 | Methylglyoxal + Glycine | 17.0 | 0.0 | 53.0 | 0.0 | 88.0 | 0.0 | 1.2 ± 0.4 | 18.4 ± 3.1 | Liquid |
| 2 | Glycolaldehyde + Methylamine | 15.0 | 0.0 | 19.0 | 0.0 | 47.0 | 0.0 | 1.2 ± 0.4 | 17.9 ± 2.4 | Liquid |
| 3 | Glyoxal + Ammonium Sulfate | 30.0 | 0.0 | 9.0 | 0.0 | 35.0 | 0.0 | 1.3 ± 0.6 | 14.1 ± 3.5 | Liquid |




| | Miscellaneous non-biological | | | | | | | | |
|---|---|---|---|---|---|---|---|---|---|
| 1 | Laboratory wipes | 112.0 | 30.6 | 54.0 | 15.2 | 47.0 | 15.4 | 3.6v5.7 | 16.4 ± 14.4 | Rubbed material over inlet |
| 2 | Cotton t-shirt (white) | 567.0 | 34.9 | 145.0 | 16.1 | 139.0 | 16.4 | 4.9 ± 4.7 | 23.5 ± 16.2 | |
| 3 | Cotton t-shirt (black) | 56.0 | 13.5 | 22.0 | 1.7 | 34.0 | 1.5 | 2.7 ± 4.0 | 17.6 ± 14.8 | |




## 9. Figures

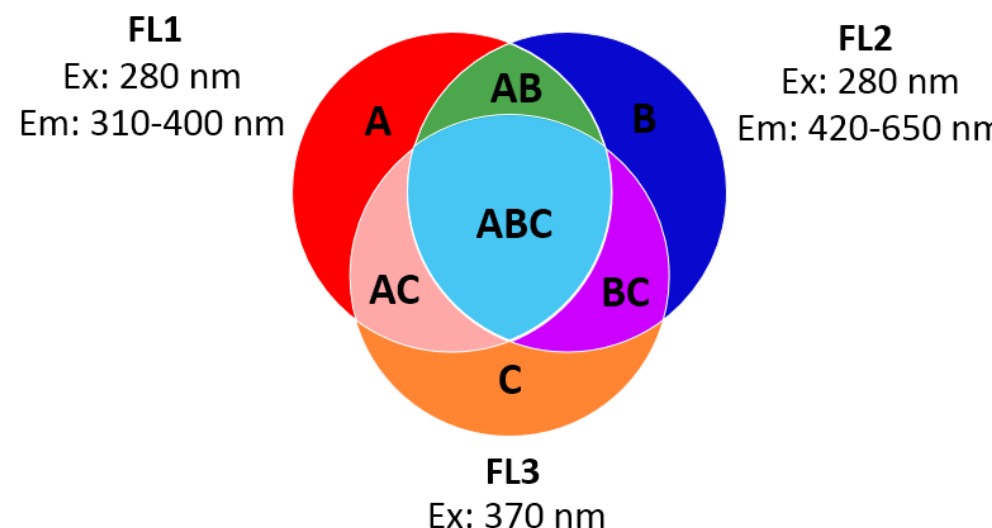

**1135**

Figure 1. Particle type classification, as introduced by introduced by Perring et al. (2015). Large circles each represent one fluorescence channel (FL1, FL2, FL3). Colored zones represent particle types that each exhibit fluorescence in one, two, or three channels.



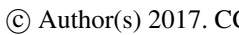


Figure 2. Representations including 4 of the 5 parameters recorded by the WIBS: FL1, FL2, FL3,
and particle size. Biological material types (a-c), bio-fluorophores (d-f), and non-biological
particle types (g-i). Data points represent median values. Gray ovals are shadows (cast directly
downward onto the bottom plane) included to help reader with 3-D representation. Tags in (d)
and (g) used to differentiate particles of specific importance within text.





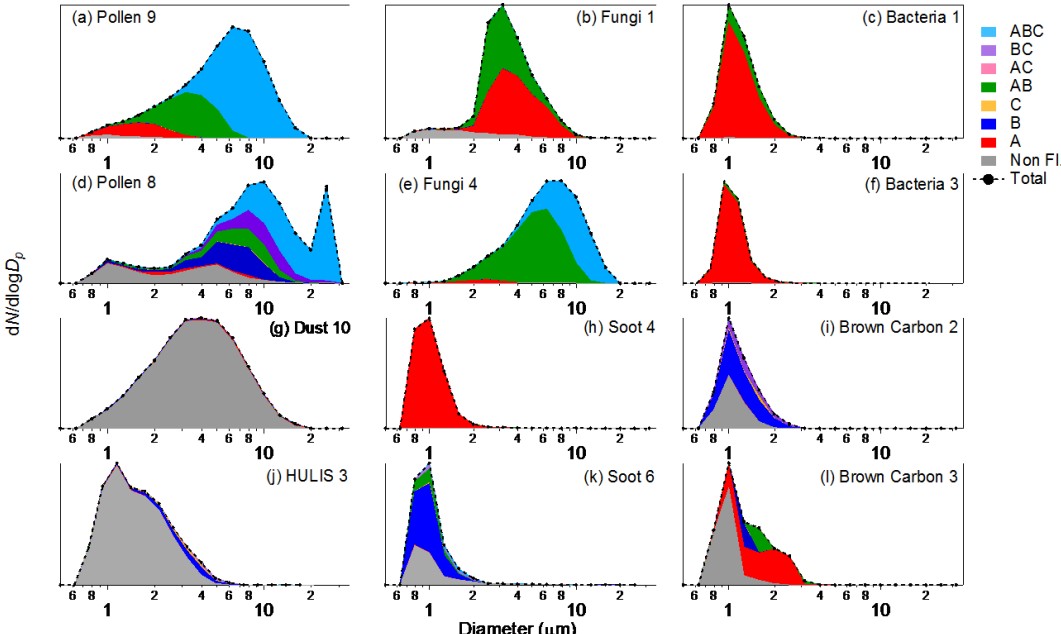


Figure 3. Stacked particle type size distributions including particle type classification, as
introduced by introduced by Perring et al. (2015) using FT + 3σ threshold definition. Examples
of each material type were selected to show general trends from larger pool of samples. Soot 4
(h) as an example of combustion soot and Soot 6 (wood smoke) as an example of smoke aerosol.



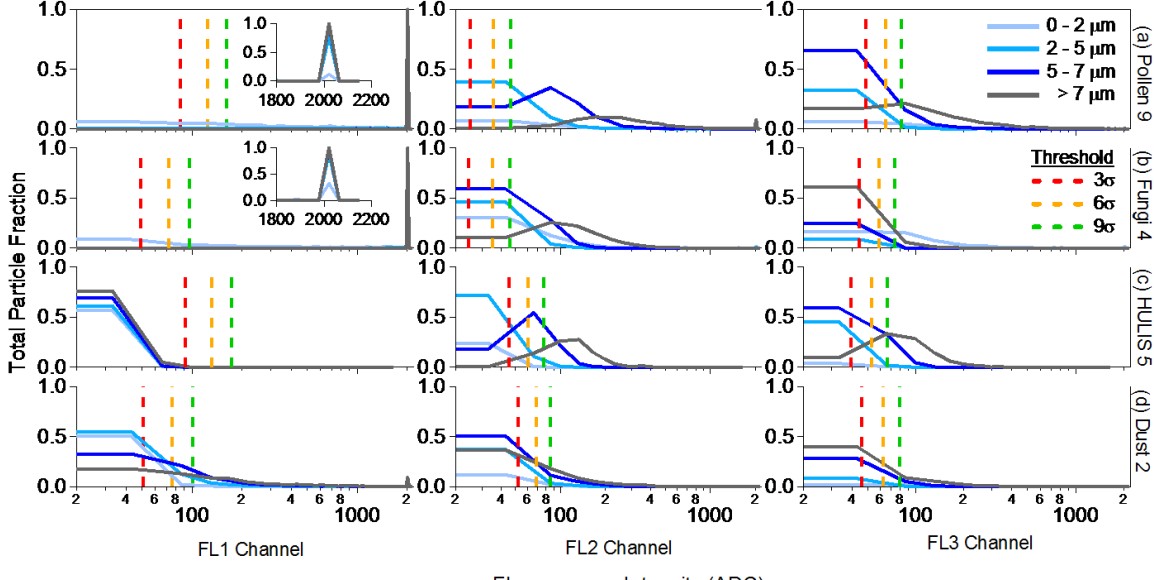

1150

1151

Figure 4. Relative fraction of fluorescent particles versus fluorescence intensity in analog-to-digital counts (ADC) for each channel. Particles are binned into 4 different size ranges (trace colors). Vertical lines indicate three thresholding definitions. Insets shown for particles that exhibit fluorescence saturation characteristics.

1156



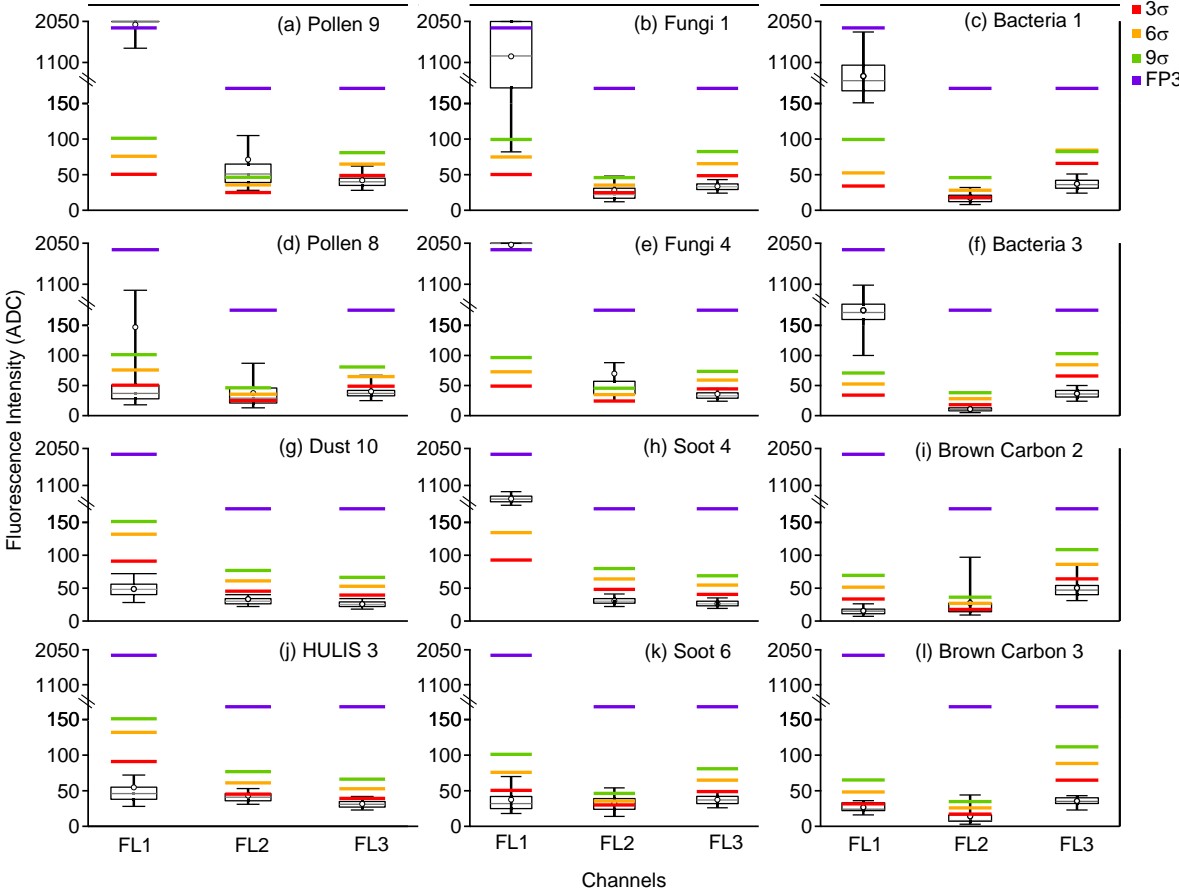

1157

Figure 5. Box whisker plots showing statistical distributions of fluorescence intensity in analog-
to-digital counts (ADC) in each channel. Averages are limited to particles in the size range 3.5-
4.0 µm for pollen, fungal spore, HULIS, and dust samples and in the range 1.0-1.5 µm for
bacteria, brown carbon, and soot samples. Horizontal bars associated with each box-whisker
show four separate threshold levels.





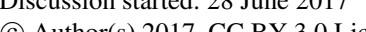

1163

1164

Figure 6. Fraction of particle number exhibiting fluorescent in a given channel versus particle diameter for various material types for four different thresholds definitions. Data markers shown only when disambiguation of traces is necessary. Brown carbon sample denoted by BrC.





Figure 7. Stacked particle type size distributions for representative particle classes shown using four separate thresholding strategies. NF+ particle type (right-most column) represents particles that exceed the FL2 and/or FL3 upper bound of the Wright et al. (2014) FP3 definition and that are therefore considered as one set of "non-fluorescent" particles by that definition. Legend above top rows indicate threshold definition used.



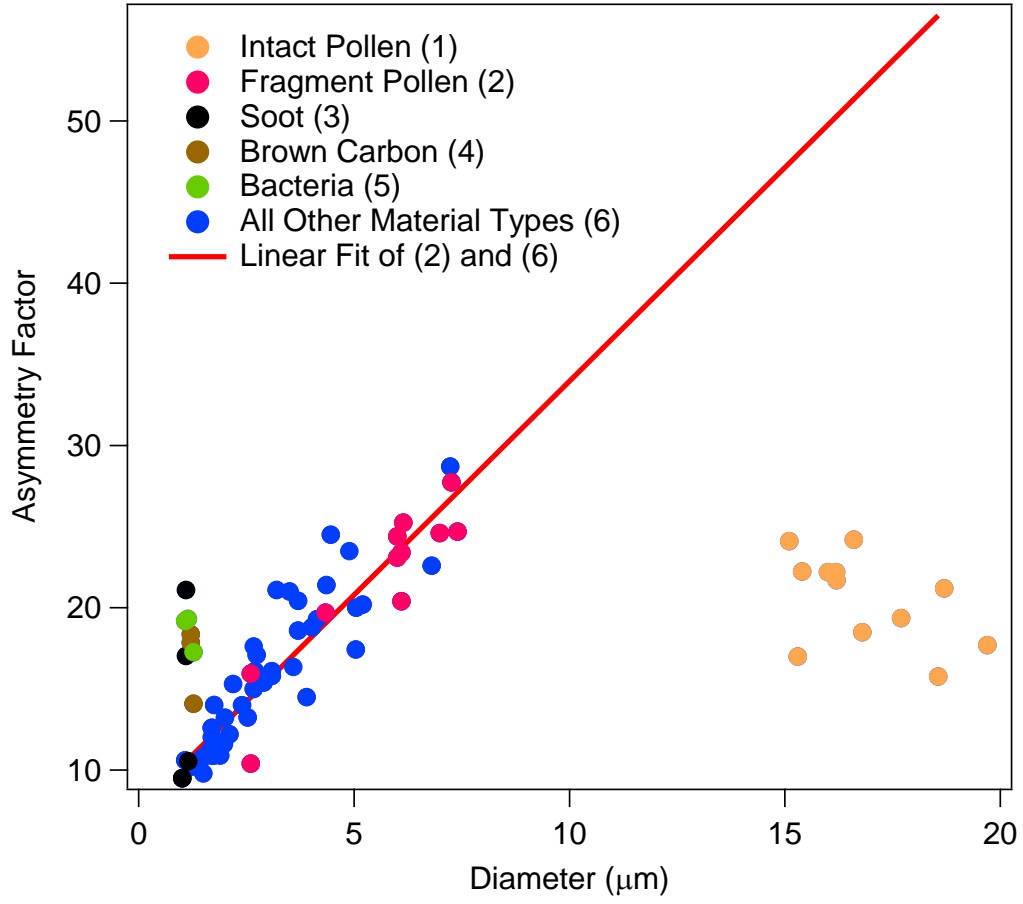

1181

Figure 8. Median values of particle asymmetry factor versus particle size for all particle types
analyzed. Fitted linear regression shown, with equation y = 2.63x +7.64 and $R^2$ = 0.87. Linear
regression analysis was done for samples pooled from the categories of Fragmented Pollen (2)
and All Other Material Types (6).