# Peer review of "Systematic Characterization and Fluorescence Threshold Strategies for the Wideband Integrated"

_Atmospheric Measurement Techniques, 2017_

## Referee Comment (RC1) · Anonymous Referee #2 · 22 Jul 2017

The manuscript is very well written and I believe of great relevance to the bioaerosol scientific community. The authors present very interesting and novel work testing a Light induced fluorescence (LIF) instrument (WIBS-4A) whilst attempting to display the data in new ways. Thus I believe the paper should be published upon the correction of some minor technical/specific issues discussed below.

Specific/technical comments:

L196 I believe that this line is misleading, while a value of 0 does indicate a particle is

a perfect sphere values just above this do not indicate that they are rod-like as directed by the sentence "Whereas larger AF values greater than 0 and less than 100, indicate rod-like particles"

What is the average/median AF value seen for PSL for instance? I doubt they are seen to be 0. Values increasing towards 100 do indicate an increasing rod-like morphology however Indeed placement of the AF values of the PSL sphere in table one would be useful.

L 302 What is a blade of air? Blast perhaps?

L 337 What was considered sufficiently fine?

Table 2 Pyrdoxine particle 7 in Biofluorophores has no number in the saturated column

Were there any issues with contamination whilst using a NAD?

L555 Are intact pollen not counted? Or do they saturate the sizing detector and are thus mis-sized?

L560-3 Given that the pollen are disrupted, they now have the intine of the pollen exposed. Thus is it this rather than the fraction of the pollen that is radiated the most important?

L609 should the line say "adds either A and C" rather than "adds either B and C"

L647 tryptophan does not appear to follow A –> BC –> ABC pathway from visual inspection of the associated graph.

Similarly in the discussion of the pathways for riboflavin the particles appear to have either B or C character to start with before gaining the required character to become BC. The pathway you describe does not suggest this. It suggests that particles pass from B to C to BC

---

## Referee Comment (RC2) · AE Perring (Referee) · 10 Aug 2017

Savage review, Aug. 2017

This manuscript presents a very large set of laboratory observations of different kinds of fluorescent aerosol (both biological and non-biological) using a WIBS 4A, presented in the context of a recent analysis framework. The authors use this dataset to evaluate the ability of the WIBS to detect a variety of biological aerosol, to characterize the observed response in a particular instrument and to make recommendations for excluding

common interferents. They have also extended the utility of the analysis framework by systematically investigating the effect of size on the fluorescence response for a given bioaerosol population and have additionally evaluated the performance of the asymmetry factor parameter, an output which is often used but which is of unknown value in distinguishing different types of particles.

The paper is well written and the community is sorely in need of this kind of characterization and criticial analysis of performance if we are to make robust measurements of atmospheric abundances of bioaerosol. Questions of potential interferences are one of the largest hurdles in the use of UV-LIF technologies and this paper is a valuable piece of that puzzle. I have a few comments and suggestions as outlined below for the authors to consider but I certainly recommend publication in ACP with only minor modifications.

Specific comments:

1. On p5, I'm not totally sure how you guys are doing the calibration but I think you should probably include a bit more detail. Did you just run a few sizes of PSL and then fit with a 2nd order polynomial? Was there any consideration of the expected instrumental response given Mie theory? I have run some calculations of expected response and compared that to PSLs and usually get reasonable correspondence but I'm not sure than a 2nd degree polynomial is sufficient to capture the expected shape of the response. Admittedly any differences are likely at the larger sizes and probably don't impact the results much but size is one of the parameters that is used heavily and there seems to be wide variability in how it is treated. Most critically the size you are reporting is not simply the size the WIBS reports based on its internal calibration but is instead based on the observed peak heights and calibrated by you using multiple PSL sizes. I think this point could be made clearer as many WIBS users seem to still use the WIBS internal calibration, simply checked periodically with one size of PSLs. 2nd order polynomial extrapolation to larger sizes than are represented by PSLs are an additional uncertainty.

2. Can you include a statement and/or reference for how representative these chemically-produced "brown carbon" compounds are of atmospheric brown carbon? This may be addressed in the Powelson reference and you do discuss it a bit later in the paper, however it would be useful to have some discussion of this in the methods section when brown carbon is introduced. I.e., we know it's a surrogate but it's the best option we have. We expect the absorption spectrum is similar but the cross section is different by. . .

3. Initially it took me a while to figure out what you meant in the text and figures by "miscellaneous particles". Although the samples are delineated in the table, it might be better to relabel "miscellaneous particles" as "common household fibers" or something more descriptive for ease of reading.

4. I think it is worth explicitly noting somewhere in this manuscript that all of the populations sampled are fresh samples and we do not know how atmospheric aging would impact our ability to detect ambient bioaerosols. It is a necessary benchmark to understand what the fresh emissions would look like however we do not know how the fraction of particles detected would change over time so this may not perfectly reflect (would be a best case scenario of?) our ability to detect ambient particles.

5. I think the nuances of what you are seeing with the dust is critically useful and I would like to see a bit more context for these numbers and more detailed discussion of the different samples rather than lumping them all into a "dust" category. The expectation is that dust, by number, is much more abundant than bioaerosols such that, even if only 1% of a certain population of dust is misidentified, it could be a huge number relative to the abundance of bioaerosol. I suggest expanding the discussion of the dust to include where these dusts are from and whether you have any idea about how abundant these different kinds of dust are in the atmosphere. Is it possible at this stage to put bounds on how much dust may impact WIBS measurements in different environments?

6. The suite of particles investigated is impressive and I can appreciate that it is not

reasonable to discuss each individual particle type in detail. However, similar to the above comment, I think the current discussion is a little bit too case-study oriented and would benefit from a bit more distillation/bigger picture. I found myself wondering how representative Hulis 5 and the 15% fluorescent dust particles are of those populations. This is already addressed somewhat but I recommend expanding the discussion or possibly adding a section specifically about implications of known interferences on ambient measurements.

7. It seems that these results are fairly consistent with the Hernandez et al findings except for a couple of things. First, there are a lot of non-fluorescent particles in several of the pollen samples if I'm reading the supplemental graphs correctly. This is surprising as we have always found nearly all pollen particles in a sample to be fluorescent in previous analyses (i.e. the Hernandez paper). It's a little hard to see it in the Hernandez paper but, if you add up each row in Table A1 (which shows the percentage of a given sample that showed up as a particular type), they don't quite sum to 100% and, for at least those pollen samples, we had >95% of all particles detected as fluorescent. So I am surprised to see so many pollens with a large non-fluorescent contribution here. Second, in Hernandez, the type B presentation was at most a minor (<10%) fraction of particles for a given population and even that only appeared in a handful of biological samples (for two different instruments). Here it seems that many of the pollen samples have a substantial fraction of particles manifesting as type B. This is unfortunate as it seems that type B is often also found in possible non-biological interferents. Have the authors thought about what might drive this kind of variability? I suppose it could be specific to certain pollen species, it could be instrument variability or it could be something to do with the samples or nebulization but this probably deserves a little discussion.

8. The discussion of the size dependence of fluorescence is nice. I think it would be worth double checking that there is not a size-dependence in the FL2 detector for non-fluorescent particles. I think there was a batch of bad notch filters at some point in

WIBS production that led to some bleed through of flash lamp light to that detector. This may be somewhat hard to assess given that some PSLs have a fluorescent surfactant (and thus "normal" non-fluorescent-doped PSLs will sometimes fluoresce) but it can be done with dioctyl sebacate or AmSO4 or any other non-fluorescent material (which need not be mono disperse).

9. I appreciate your discussion of the asymmetry factor and the potential problems with it. On lines 726-727 I believe you meant to say that the forward-scattering detector may not be able to reliably estimate either size or AF? I also think you could give at least a hint at your ultimate conclusion about the AF measurement in your initial discussion of this measurement and, possibly, in the abstract. On my first read-through, after seeing the AF calculation in the text and the AF values included in the table, I thought you might not examine that parameter critically. Just something along the lines of "The performance of the asymmetry factor is assessed across populations as a function of particle size."

---

## Author Response (AR1)

September 8, 2017

Dear Associate Editor Dr. Pope,

Re: Revisions of amt-2017-170 by Savage et al.

Here you will find a summary of revisions for our recently reviewed manuscript. Both referees recommended publication after relatively minor changes and comments. We have responded point-by-point to these comments and are confident that the manuscript is improved and ready for acceptance. The only substantive change to the manuscript is the addition of a few additional paragraphs of discussion adding context to the results, as requested by Referee #2.

Attached within this document you will find documents in the following order:
- Point-by-point responses to Referees #1 and 2 (copied directly from documents uploaded to AMT)
- Revised manuscript (with all changes tracked and highlighted in yellow for changes requested by referees and in green for all other minor edits)
- Manuscript supplement

With these changes we hope you will find the revised manuscript soon acceptable for publication.

Best Regards,

J. Alex Huffman, Ph.D.
Associate Professor

**Response to referee comment on amt-2017-170 by Savage et al.**

**Anonymous Referee #1**

Note regarding document formatting: black text shows original referee comment, blue text shows author response, and red text shows quoted manuscript text. Changes to manuscript text are shown as *italicized and underlined*. Bracketed comment numbers (e.g. [R1.1] ) were added for clarity. All line numbers refer to discussion/review manuscript.

General Comments: The manuscript is very well written and I believe of great relevance to the bioaerosol scientific community. The authors present very interesting and novel work testing a Light induced fluorescence (LIF) instrument (WIBS-4A) whilst attempting to display the data in new ways. Thus I believe the paper should be published upon the correction of some minor technical/specific issues discussed below.

Author response: We thank the referee for his/her positive assessment and summary.

Specific/technical comments:

[R1.1] L196 I believe that this line is misleading, while a value of 0 does indicate a particle is a perfect sphere values just above this do not indicate that they are rod-like as directed by the sentence "Whereas larger AF values greater than 0 and less than 100, indicate rod-like particles" What is the average/median AF value seen for PSL for instance? I doubt they are seen to be 0. Values increasing towards 100 do indicate an increasing rod-like morphology however Indeed placement of the AF values of the PSL sphere in table one would be useful.

[A1.1] As requested, we added median values (± standard deviation) of AF to Table 1 for PSLs.

To clarify the statement we added text in this paragraph at L198 (italicized text added):
"A perfectly spherical particle would theoretically exhibit an AF value of 0, whereas larger AF values greater than 0 and less than 100 indicate rod-like particles (Kaye et al., 1991;Gabey et al., 2010;Kaye et al., 2005). *In practice, spherical PSL particles exhibit a median AF value of ~ 5 (Table 1).* It is important to note that *the AF* parameter is not rigorously a shape factor like used in other aerosol calculations (DeCarlo et al., 2004;Zelenyuk et al., 2006) and only very roughly relates a measure of particle sphericity."

[R1.2] L 302 What is a blade of air? Blast perhaps?

[A1.2] We added text at L302 to clarify the description of the experiment.
"For each experiment, an agar plate with a mature fungal colony was sealed inside the chamber. *A thin, wide* nozzle was positioned so that *the delivered air stream approximated* a blade of air *that* approach*ed* the top of the spore colony at a shallow angle in order to eject spores into a *roughly* horizontal trajectory."

[R1.3] L 337 What was considered sufficiently fine?

[A1.3] We added clarifying text at L337:

"The setup was modified (method P2) for a small subset of samples whose solid powder was sufficiently fine to produce high number concentrations *of particles (e.g. >200 cm$^{-3}$) and that contained enough* submicron aerosol *material to* risk coating the internal flow path and damaging optical components of the instrument."

[R1.4] Table 2 Pyrdoxine particle 7 in Biofluorophores has no number in the saturated column

[A1.4] We added missing values for Pyrdoxine in Table 2.

[R1.5] Were there any issues with contamination whilst using a NAD?

[A1.5] There were no contamination issues while running NAD, but the fear of contamination was one reason we employed aerosolization method P2. Between each sample, the instrument ran pumping for about 10 min to prevent contamination. If the baseline of that ambient data collected in those 10 min was higher, other measures were taken to ensure the optical cavity was not coated.

[R1.6] L555 Are intact pollen not counted? Or do they saturate the sizing detector and are thus mis-sized?

[A1.6] Intact pollen that make it into the instrument are counted. Most pollen grains are much larger than the upper size limit of the instrument (~20 µm), however. Thus, species of pollen with large grain sizes exhibit a size mode in the WIBS near this upper size limit. (e.g. Pollen 1, 2, 5, etc.). Any particles larger than this are integrated into the largest sizing bin, which saturates the sizing detector. A clarifying sentence was added:

L557: "… upper size limit of particle collection (~20 µm as operated). *Particles larger than this limit saturate the sizing detector and are binned together into the ~20 µm bin.*"

[R1.7] L560-3 Given that the pollen are disrupted, they now have the intine of the pollen exposed. Thus is it this rather than the fraction of the pollen that is radiated the most important?

[A1.7] The intact pollen and fragmented pollen indeed present different types of material to the excitation pulses and may, therefore, present different emission properties as a result. We believe the following, existing text clarifies this point:

L557: "It is important to note that excitation pulses from the Xe flash lamps are not likely to penetrate the entirety of large pollen particles, and so emission information is likely limited to outer layers of each pollen grain. Excitation pulses can penetrate a relatively larger fraction of the smaller pollen fragments, however, meaning that the differences in observed fluorescence may arise from differences the layers of material interrogated."

[R1.8] L609 should the line say "adds either A and C" rather than "adds either B and C"

[A1.8] This was a typo. The text was modified to correct this error:

L 609: "The "pathway" of change, for Pollen 9, starts as A-type at small particle size and adds B and eventually ABC (A→AB→ABC), whereas Pollen 8 starts primarily with B-type at small particle size and separately adds either  *A* or C en route to ABC (B→AB or BC→ABC)."

[R1.9] L647 tryptophan does not appear to follow A –> BC –> ABC pathway from visual
inspection of the associated graph.

[A1.9] This was also a typo. The pathway listed for tryptophan was correct, as follows:

"For example Biofluorophore 1 (riboflavin) follows the pathway B→C→BC while
Biofluorophore 11 (tryptophan) follows the pathway A→BC *AB*→ABC."

[R1.10] Similarly in the discussion of the pathways for riboflavin the particles appear to have
either B or C character to start with before gaining the required character to become BC. The
pathway you describe does not suggest this. It suggests that particles pass from B to C to BC

[A1.10] The referee brings up a good point here. The concept of "pathway" here does not
make sense to move from B to C to BC. Instead, there is a population of B particles and a
separate population of C particles, each of which can separately move to become BC
particles as particle size increases. To clarify this, the text has been changed as follows:

L646: "For example Biofluorophore 1 (riboflavin) follows the pathway B *or* C→BC …"

**Response to referee comment on amt-2017-170 by Savage et al.**

**Referee #2: Anne Perring**

Note regarding document formatting: black text shows original referee comment, blue text shows author response, and red text shows quoted manuscript text. Changes to manuscript text are shown as *italicized and underlined*. All line numbers refer to discussion/review manuscript.

General Comments: This manuscript presents a very large set of laboratory observations of different kinds of fluorescent aerosol (both biological and non-biological) using a WIBS 4A, presented in the context of a recent analysis framework. The authors use this dataset to evaluate the ability of the WIBS to detect a variety of biological aerosol, to characterize the observed response in a particular instrument and to make recommendations for excluding common interferents. They have also extended the utility of the analysis framework by systematically investigating the effect of size on the fluorescence response for a given bioaerosol population and have additionally evaluated the performance of the asymmetry factor parameter, an output which is often used but which is of unknown value in distinguishing different types of particles. The paper is well written and the community is sorely in need of this kind of characterization and critical analysis of performance if we are to make robust measurements of atmospheric abundances of bioaerosol. Questions of potential interferences are one of the largest hurdles in the use of UV-LIF technologies and this paper is a valuable piece of that puzzle. I have a few comments and suggestions as outlined below for the authors to consider but I certainly recommend publication in ACP with only minor modifications.

Author response: We thank the referee for her positive assessment and summary.

Specific/technical comments:

[R2.1] On p5, I'm not totally sure how you guys are doing the calibration but I think you should probably include a bit more detail. Did you just run a few sizes of PSL and then fit with a 2nd order polynomial? Was there any consideration of the expected instrumental response given Mie theory? I have run some calculations of expected response and compared that to PSLs and usually get reasonable correspondence but I'm not sure than a 2nd degree polynomial is sufficient to capture the expected shape of the response. Admittedly any differences are likely at the larger sizes and probably don't impact the results much but size is one of the parameters that is used heavily and there seems to be wide variability in how it is treated. Most critically the size you are reporting is not simply the size the WIBS reports based on its internal calibration but is instead based on the observed peak heights and calibrated by you using multiple PSL sizes. I think this point could be made clearer as many WIBS users seem to still use the WIBS internal calibration, simply checked periodically with one size of PSLs. 2nd order polynomial extrapolation to larger sizes than are represented by PSLs are an additional uncertainty.

[A2.1] The referee introduces an important point that we didn't explicitly discuss in the manuscript. In particular, we agree that particle sizing reported by the WIBS instrument is critically erroneous if not properly calibrated. To clearly introduce this concept and the method by which we calibrated particle size, the following text was added to Section 2.2:

"*The particle size reported by the internal WIBS calibration introduces significant sizing errors and critically needs to be calibrated before analyzing or reporting particle size. Particle size calibration was achieved here by using a one-time 27-point calibration curve generated using non-fluorescent PSLs ranging in size from 0.36 to 15 µm. This calibration involved several steps. For each physical sample, approximately 1,000 to 10,000 individual particles were analyzed using the WIBS (several minutes of collection). Data collected for each samples was analyzed by plotting a histogram of the side scatter response reported in the raw data files (FL2_sctpk). A Gaussian curve was fitted to the most prominent mode in the distribution. The median value of the fitted peak for observed side scatter was then plotted against the physical diameter (as reported on the bottle) for each PSL sample. A $2^{nd}$ degree polynomial function was fitted to this curve to create the calibration equation that was used on all laboratory data used here. The calibration between observed particle size and physical diameter may be affected by wiggles in the optical scattering relationship suggested by Mie theory. These theoretical considerations were not used for the calibrations reported here, and so uncertainties in reported size are expected to increase at larger diameters.*

*Following the one-time 27-point calibration, the particle sizing response was checked periodically using a 5-point calibration. The responses of these calibration checks were within one standard deviation unit of each other and so the more comprehensive calibration was always used. These quicker checks were performed using* non-fluorescent PSLs (Polysciences, Inc., Pennsylvania), including 0.51 µm (part number 07307), 0.99 µm. (07310), 1.93 µm (19814), 3.0 µm (17134), and 4.52 µm (17135)."

[R2.2] Can you include a statement and/or reference for how representative these chemically-produced "brown carbon" compounds are of atmospheric brown carbon? This may be addressed in the Powelson reference and you do discuss it a bit later in the paper, however it would be useful to have some discussion of this in the methods section when brown carbon is introduced. I.e., we know it's a surrogate but it's the best option we have. We expect the absorption spectrum is similar but the cross section is different by…

[A2.2] Indeed, there are many different pathways to brown carbon formation in the atmosphere. We chose to utilize methods published by Powelson et al. (2014) primarily because the experiments were more easily achievable due to their bulk-phase nature and because we did not need to find access to a reaction flow-tube. Small, water soluble carbonyl compounds such as methylglyoxal, glycolaldehyde and glyoxal can undergo Maillard-type browning reactions or aldol condensation reactions in the presence of ammonium salts, amino acids (glycine) or primary amines (methylamine), like those reagents used in this study. Table 1 in the Powelson et al. (2014) reference reports atmospheric concentrations (in both cloud and aerosol) for each reagent used here. In the last paragraph of their paper the authors also present a short analysis of global emissions of these compounds, concluding in the last line of the paper that "because of lower MAC
[mass absorption coefficients] values for products of aldehyde-amine-AS browning
reactions, they are likely responsible for <10% of light absorption by atmospheric brown
carbon." We felt these details were beyond the scope of relevance for our manuscript,
but have added a few sentences of context to the methods (Section 3.1.2) as requested.

L271: "*These reactions were chosen, because the reaction products were achievable*
*using bulk-phase aqueous chemistry and did not require more complex laboratory*
*infrastructure. They represent three examples of reactions possible in cloud-water using*
*small, water-soluble carbonyl compounds mixed with either ammonium sulfate or a*
*primary amine (Powelson et al., 2014). A large number of reaction pathways exist to*
*produce atmospheric brown carbon, however, and the products analyzed here are*
*intended primarily to introduce the possible importance of brown carbon droplets and*
*coatings to fluorescence-based aerosol detection (Huffman et al., 2012).*"

Reference: Powelson, M. H., Espelien, B. M., Hawkins, L. N., Galloway, M. M., and De
Haan, D. O.: Brown Carbon Formation by Aqueous-Phase Carbonyl Compound
Reactions with Amines and Ammonium Sulfate, Environmental Science & Technology,
48, 985-993, 10.1021/es4038325, 2014.

[R2.3] Initially it took me a while to figure out what you meant in the text and figures by
"miscellaneous particles". Although the samples are delineated in the table, it might be better to
relabel "miscellaneous particles" as "common household fibers" or something more descriptive
for ease of reading.

[A2.3] This is a good idea and we have changed "miscellaneous particles" to "common
household fibers" in all places that it occurred in the manuscript text, figures, and
supplement.

[R2.4] I think it is worth explicitly noting somewhere in this manuscript that all of the
populations sampled are fresh samples and we do not know how atmospheric aging would
impact our ability to detect ambient bioaerosols. It is a necessary benchmark to understand what
the fresh emissions would look like however we do not know how the fraction of particles
detected would change over time so this may not perfectly reflect (would be a best case scenario
of?) our ability to detect ambient particles.

[A2.4] We have added the following text after L267:
*"It is important to note that all particle types analyzed here essentially represent "fresh"*
*emissions. It is unclear how atmospheric aging might impact their surface chemical*
*properties or how their observed fluorescence properties might evolve over time."*

[R2.5] I think the nuances of what you are seeing with the dust is critically useful and I would
like to see a bit more context for these numbers and more detailed discussion of the different
samples rather than lumping them all into a "dust" category. The expectation is that dust, by
number, is much more abundant than bioaerosols such that, even if only 1% of a certain
population of dust is misidentified, it could be a huge number relative to the abundance of bioaerosol. I suggest expanding the discussion of the dust to include where these dusts are from and whether you have any idea about how abundant these different kinds of dust are in the atmosphere. Is it possible at this stage to put bounds on how much dust may impact WIBS measurements in different environments?

[A2.5] All dust samples were generously loaned from a collection in the Department of Geology and Earth Science in the School of Earth and Environmental Sciences at the University of Manchester, and we were not able to investigate details regarding atmospheric concentrations and geographic trends associated with each.

The referee's question about constraining the importance of weakly fluorescent non-biological material is an important point of discussion, but also very complicated. Prompted by the important comment we included a simple analysis along with a relatively detailed additional paragraph suggesting the general scenarios that could increase quantitative uncertainties and the impact these may have on conclusions drawn about an ambient air mass. The following text was inserted at L795:

[revised manuscript text omitted]

[R2.6] The suite of particles investigated is impressive and I can appreciate that it is not
reasonable to discuss each individual particle type in detail. However, similar to the above
comment, I think the current discussion is a little bit too case-study oriented and would benefit
from a bit more distillation/bigger picture. I found myself wondering how representative Hulis 5
and the 15% fluorescent dust particles are of those populations. This is already addressed
somewhat but I recommend expanding the discussion or possibly adding a section specifically
about implications of known interferences on ambient measurements.

[A2.6] Textual context was added to the manuscript as a part of response [R2.5].
Additionally, we investigated the properties of HULIS 5, which was presented within the
manuscript as an outlier in terms of high fluorescence, as suggested by the referee. This
material is indeed not expected to be a common type of material one would expect to see
in the atmosphere, as discussed in the text added below (after L522):

*"HULIS 5 is a fulvic acid collected from a eutrophic, saline coastal pond in Antarctica.*
*The collection site lacks the presence of terrestrial vegetation, and therefore all dissolved*
*organic material present originates from microbes. HULIS 5, therefore, is not expected*
*to be representative of soil-derived HULIS present in atmospheric samples in most areas*
*of the world. We present the properties of this material as an example of relatively highly*

*fluorescing, non-biological aerosol types that could theoretically occur, but without*
*comment about its relative importance or abundance."*

The following text was modified at L685:
"As a 'worst case' scenario, HULIS 5 shows ca. 60% of particles to be fluorescent using
the 3σ threshold, *but this material is unlikely to be representative of commonly observed*
*soil HULIS, as discussed above."*

The following text was modified at L785:
"It is important to note that HULIS 5 was one of a large number of analyzed particle
types and in the minority of HULIS types, however, and it is *unlikely that this microbe-*
*derived material*  *would be observed*
in a given ambient air mass *at most locations.* More studies may be
required to sample dusts, HULIS types, soot and smoke, brown organic carbon materials,
and various coatings in different real-world settings *and at various stages of aging* to
better understand how specific aerosol types may contribute to UV-LIF interpretation at a
given study location."

[R2.7] It seems that these results are fairly consistent with the Hernandez et al findings except
for a couple of things. First, there are a lot of non-fluorescent particles in several of the pollen
samples if I'm reading the supplemental graphs correctly. This is surprising as we have always
found nearly all pollen particles in a sample to be fluorescent in previous analyses (i.e. the
Hernandez paper). It's a little hard to see it in the Hernandez paper but, if you add up each row in
Table A1 (which shows the percentage of a given sample that showed up as a particular type),
they don't quite sum to 100% and, for at least those pollen samples, we had >95% of all particles
detected as fluorescent. So I am surprised to see so many pollens with a large non-fluorescent
contribution here. Second, in Hernandez, the type B presentation was at most a minor (<10%)
fraction of particles for a given population and even that only appeared in a handful of biological
samples (for two different instruments). Here it seems that many of the pollen samples have a
substantial fraction of particles manifesting as type B. This is unfortunate as it seems that type B
is often also found in possible non-biological interferents. Have the authors thought about what
might drive this kind of variability? I suppose it could be specific to certain pollen species, it
could be instrument variability or it could be something to do with the samples or nebulization
but this probably deserves a little discussion.

[A2.7] It is an interesting comment that the fraction of pollen grains exhibiting
fluorescence as reported by the Hernandez et al. paper was e.g. >95%, whereas more
pollen species are shown here with higher non-fluorescent fractions. Most pollen species
were used only in either the Hernandez et al. paper or our work, but not both. *Phleum*
*pratense* is the only exception, used in both studies, and it interestingly shows similar
non-fluorescent fractions of ~2% or less in both manuscripts. Similarly, the fraction of
*Phleum pratense* shown in Figure 2 of Hernandez et al. (visually) shows approximately
95% of particles to have B-type properties. This fraction is similar to the fraction we
report (i.e. Figure 3a). This could indicate a higher degree of instrumental agreement than
initially obvious and that observed differences in fluorescent properties are influenced
heavily by the choice of pollen grains analyzed in both studies.

That said, there are clear reasons one would expect instrument to show different patterns to separately aerosolized pollen. For example:

   (1) The conditions for pollen growth and biological state may be different, given that the pollen came from different distributors. The storage conditions, age, and aerosolization processes were also different and could impact the chemical and physical states of the material as well as the fraction of pollen grains that fractured before analysis.

   (2) The observed differences in fluorescent properties can also be heavily influenced by instrument properties. For example, instrument gains can be set differently in each instrument. It may be that our FL2 detector has higher sensitivity, resulting in more B fraction particles.

It is unclear how all these factors might combine to quantitatively compare the minor differences between observations. The most reliable answer to improve differences in results would be to perform similar laboratory measurements with collocated instruments, which we suggest could be important to the community. Beyond this, it is becoming increasingly clear that calibrating different WIBS instruments based on an absolute fluorescence standard is critically important. Work like the referee's recent paper (Robinson et al., 2016) will help solve similar conundrums in the future.

[R2.8] The discussion of the size dependence of fluorescence is nice. I think it would be worth double checking that there is not a size-dependence in the FL2 detector for non-fluorescent particles. I think there was a batch of bad notch filters at some point in WIBS production that led to some bleed through of flash lamp light to that detector. This may be somewhat hard to assess given that some PSLs have a fluorescent surfactant (and thus "normal" non-fluorescent-doped PSLs will sometimes fluoresce) but it can be done with dioctyl sebacate or AmSO4 or any other non-fluorescent material (which need not be mono disperse).

[A2.8] Based on the referee's suggestion, we looked into the size-dependence of the FL2 detector, as shown below. Histogram plots of fluorescence intensity in each fluorescence channel were created for each PSL sample, and Gaussians fits were applied to each mode present (3 peaks in Figure R.1). To determine whether there was a particle size dependence on the FL2 detector, four pieces of information were extracted from each histogram and plotted as a function of PSL particle diameter (Fig. R.2). Figures R.2A and B show the relationship of the median intensity of the two non-saturating modes from the histogram. Figure R.3-C shows the percent of particles that saturated the FL2 detector, and Figure R.3-D shows the median fluorescence intensity of all the data. Non-fluorescent PSLs ranging in size from 0.3 – 15 μm in size were plotted in Figure R.2, the two colors representing size calibrations from two separate occasions.

The two data sets show no obvious size correlation for peak 1 or peak 2 present in the FL2 channel, seen as essentially a flat relationship in Figure R.2A and R.2B. If there was a size dependence on the FL2 detector one would expect an increase in FL2 intensity as a function of particle size increases. There is an increase in percent FL2 saturation values for PSLs between ~1 and 4 μm, but only to a total of approximately 1.5% (Fig. R.2C).

Finally, overall median values for the FL2 intensity also do not show a size dependence correlation.

Based on this follow-up analysis we conclude that there was no obvious trend between the measurements at the FL2 detector and particle size. This suggests that bleed through from the flash lamp was not present in this case, and so it is unlikely that the instrument is affected by any possible bad notch filters. This suggestion was an excellent one to consider, however, and we suggest that other WIBS users be aware of this possible problem and check their instrument(s) in a similar fashion.

[Figure]

Figure R.1: Histogram of FL2 responses shows multiple fluorescent modes for these 10 um PSLs.

[Figure]

Figure R.2: (A) FL2 intensity vs. diameter for peak 1, (B) FL2 intensity vs. diameter for peak 2, percent saturation in FL2 channel vs. diameter and (C) median fluorescence intensity vs. diameter.

[R2.9] I appreciate your discussion of the asymmetry factor and the potential problems with it. On lines 726-727 I believe you meant to say that the forward-scattering detector may not be able to reliably estimate either size or AF? I also think you could give at least a hint at your ultimate conclusion about the AF measurement in your initial discussion of this measurement and, possibly, in the abstract. On my first read-through, after seeing the AF calculation in the text and the AF values included in the table, I thought you might not examine that parameter critically. Just something along the lines of "The performance of the asymmetry factor is assessed across populations as a function of particle size."

[A2.9] We changed L728:

[revised manuscript text omitted]

*ATCC: American Type Culture Collection
** University of Manchester – School of Earth and Environmental Sciences
*** International Humic Substance Society

[Figure]

Figure S1. Schematic diagram of home-built chamber for the aerosolization of fungal spores.

[Figure]

Figure S2. Impacted pollen (*Olea europaea*) images collected with an AmScope camera (MU800, AmScope) with an objective lens with 40x magnification. (a) Not stirred (b-d) Stirred.

[Figure]

Figure S3. Fluorescence intensity histogram of FL1 for *Aspergillus niger* (Fungi 2). One broad
mode extending from 0-2000 analog-to-digital counts (ADC) and a second mode showing
detector saturation at ~2047 ADC.

[Figure]

Figure S4A. Stacked particle type size distributions of pollen using FT + 3σ threshold

[Figure]

Figure S4B. Stacked particle type size distributions of pollen using FT + 9σ threshold

[Figure]

Figure S4C. Stacked particle type size distributions of fungal spores using FT + 3σ threshold

[Figure]

Figure S4D. Stacked particle type size distributions of fungal spores using FT + 9σ threshold

[Figure]

Figure S4E. Stacked particle type size distributions of bacteria using FT + 3σ threshold

[Figure]

Figure S4F. Stacked particle type size distributions of bacteria using FT + 9σ threshold

[Figure]

Figure S4G. Stacked particle type size distributions of biofluorophores using FT + 3σ  threshold

[Figure]

Figure S4H. Stacked particle type size distributions of biofluorophores using FT + 9σ  threshold

[Figure]

Figure S4I. Stacked particle type size distributions of dust using FT + 3σ threshold

[Figure]

Figure S4J. Stacked particle type size distributions of dust using FT + 9σ threshold

[Figure]

Figure S4K. Stacked particle type size distributions of HULIS using FT + 3σ threshold

[Figure]

Figure S4L. Stacked particle type size distributions of HULIS using FT + 9σ threshold

[Figure]

Figure S4M. Stacked particle type size distributions of PAHs using FT + 3σ threshold

[Figure]

Figure S4N. Stacked particle type size distributions of PAHs using FT + 9σ threshold

[Figure]

Figure S4O. Stacked particle type size distributions of soot using FT + 3σ threshold

[Figure]

Figure S4P. Stacked particle type size distributions of soot using FT + 9σ threshold

[Figure]

Figure S4Q. Stacked particle type size distributions of brown carbon (BrC) using FT + 3σ threshold

[Figure]

Figure S4R. Stacked particle type size distributions of brown carbon (BrC) using FT + 9σ threshold

[Figure]

Figure S4S. Stacked particle type size distributions of miscellaneous samples using FT + 3σ threshold

[Figure]

Figure S4T. Stacked particle type size distributions of miscellaneous samples using FT + 9σ threshold